



# An urban agglomeration effect on surface UV doses: Comparison of the Brewer measurements in Warsaw and at Belsk, Poland, for the period 2013-2015

Agnieszka E. Czerwińska[1], Janusz W. Krzyścin[1], Janusz Jarosławski[1]

[1] Institute of Geophysics, Polish Academy of Sciences, Warsaw, 01452, Poland

*Correspondence to*: Agnieszka E. Czerwińska (aczerwinska@igf.edu.pl)

**Abstract.** The specific aerosols and cloud properties over large urban regions seem to generate an island, similar to the well-know city heat island, leading to lower UV radiation intensity compared to the surrounding cleaner areas, thus creating a shield against excessive human exposure to the UV radiation. The present study focuses on differences in the erythemal and

UV-A1 (340-400 nm) doses measured by the Brewer spectrophotometers in Warsaw (52.3ºN, 21.0ºE) and at Belsk (51.8ºN, 20.8ºE), which is located in a rural region at a distance of about 60 km in the south-west direction from the city. The ratio between erythemal and UV-A1 partly daily doses, obtained during all-sky and cloudless-sky conditions in the period May 2013-December 2015, are analyzed to infer specific cloud and aerosol forcing on the surface UV doses over Warsaw. Radiative model simulations are carried out to assess impact of the Warsaw-Belsk differences in total ozone, geographical

location and albedo, on the mean ratio between the doses. Higher surface albedo over the city compensates the effect of solar elevation differences due to latitude differences as the mean total ozone values appear almost the same over both sites. It is found that urban agglomeration induced 8% and 5% attenuation of the erythemal and UV-A1 doses, respectively, which could be caused by larger aerosol absorption. It appears that a slightly increased optical depth of the urban aerosols and properties of clouds generated over Warsaw are less important for the UV attenuation. In this work we are showing that the

higher city surface albedo compensates for the solar UV attenuation caused by urban aerosol load in the city of Warsaw.





## 1 Introduction

Excessive exposure to the ultraviolet radiation (UVR) reaching the Earth's surface has a detrimental impact on the human health. Overexposure to UV-B radiation (290-315nm) can cause erythema (redness of the skin), DNA and cellular

damage (due to generation of reactive oxygen species), and immunosupression. Longer UV wavelengths, UV-A (315-400nm), can be cancerogenetic but also responsible for photoaging, and various eye diseases, including cataract. Both overexposure to UV-B and UV-A could lead to increased risks of cutaneous melanoma, non-melanoma skin cancers, and various health problems (e.g. Marionnet et al., 2014; Greinert et al., 2015). While UV-B is strongly depended on the latitude and thickness of the ozone layer, UVA, especially UV-A1, the so-called long-wave UV-A (340-400nm), is ozone

independent, more intense, and less variable with latitude (Sabziparvar et al., 1999). The absorption by $SO_2$ (in the UV-B range) and $NO_2$ (mostly in the UV-A range) is important for the surface UV attenuation only in extreme concentrations of such gases. The surface intensity of UV depends significantly on properties of clouds and aerosols.The negative trends in these variables, found over many of the northern hemisphere midlatitudinal sites in the 1989s and 1990s, lead to increases of both the UV-B and UV-A irradiance (e.g. Krzyścin et al., 2011; Zerefos et al., 2012; De Bock et al., 2014).

An attenuation of the incoming solar radiation seems to be higher over the large urban agglomeration relative to the surrounding rural areas due to the excessive light scattering and absorption by the anthropogenic aerosols. Papayannis et al. (1998) found differences between UV cloudless-sky irradiances measured over Athens and its suburbs. In Athens the concentration of atmospheric aerosols was higher than at the outskirts site. The erythemal irradiance at the centre of Athens was 30% lower than at the suburbs with similar values of total ozone ($TO_3$) for days with increased pollution in the air.

Similar difference was noticed on the basis of the numerical simulations of UV-B irradiance with input from measurements of the $TO_3$ and aerosols optical depth (AOD) by the Brewer spectrophotometer (BS) at the outskirts of Athens. Acosta and Evans (2000) measured the erythemal irradiances in the centre and suburbs of Mexico City in the period 1994-1995. During the winter, the erythemal irradiance was 9% greater at the suburbs than in the centre of Mexico, while during the summer, the recorded values were up to 43% greater (the mean value was 21%). Corr et al. (2009) found for Mexico City aerosols

enhanced absorption at UV wavelengths with single scattering albedo (SSA) in the range 0.7-0.85. Even larger attenuation of the UVR due to atmospheric aerosols of ~ 60% was reported in Guangzhou, China, in the dry season from October up to January (Deng et al., 2012). Kazadzis et al. (2009a) found that at some cloudless days, differences in AOD among three sites (urban, rural and industrial area) located in Thessaloniki and at the outskirts of the city can cause up to 20% differences in UV irradiance. Meleti et al. (2011) and Fountoulakis et al. (2016) noticed that the surface UVR measured in Thessaloniki

may be sensitive to another characteristic of the atmospheric aerosols, the single scattering albedo of aerosols, which may counteract the effects of AOD changes there.



The atmosphere over Poland is one of the most dust polluted in Europe. The PM10 and PM2.5 levels measured in Warsaw, as well as in most larger cities in Poland, exceed the tolerable PM limit many times during the year (Polish Ministry of Environment, 2012). However, Zawadzka et al. (2013) analyzed the measurements taken by the Microtops II and CIMEL sunphotometer and stated, that a small positive bias for AOD at 500 nm between Warsaw and a rural site (Belsk),

which is in ~60 km distance from the city in the south-west direction, was not larger than 0.02, whereas for greater values of wind velocity it reached 0.04. The bias calculated from satellite measurements with the MODIS (Moderate Resolution Imaging Spectroradiometer) was ~0.05. The authors did not claim any significant differences in the Angström parameters between the sites for the visible ranges, so it could be hypothesized, that AOD values in UV range also differ only slightly.

The geophysical variables possibly affecting the differences in the ground level of the surface UVR between a city
and remote sites are: clouds, aerosols, and surface albedo. It seems possible that large urban agglomeration could produce specific cloud properties (due to heat island effect and creation of a specific cloud condensation nuclei consisting of urban aerosols), higher loading of aerosols, and higher albedo. The working hypothesis is that the Warsaw agglomeration produces a kind of shield against the incoming UV radiation. We will strive to support (or disprove) the hypothesis by comparing the erythemal and UV-A1 radiation measurements by BSs in Warsaw and at Belsk for the period May 2013-December 2015.

## 15   2     Methodology

Monitoring of the UV spectra by BS is carried out by the Institute of Geophysics, Polish Academy of Sciences (IGF PAS), at the Central Geophysical Observatory Belsk since 1992 by the single monochromator BS No. 64, and in Warsaw since 2013 by the double monochromator BS No. 207 installed on the roof of IGF PAS main building. Previously, BS 207 was working at Belsk (2010-2013). Comparison of BS No. 64 and BS No. 207 for that period will allow us to assess the
20 differences between the measured UV doses due to the instrumental differences. In the middle of 2013 BS 207 was moved to Warsaw.

The present study focuses on differences in the erythemal and UV-A1 (340-400 nm) doses measured by BSs in Warsaw (52.3ºN, 21.0ºE, 130 m amsl) and at Belsk (51.8ºN, 20.8ºE, 190 m amsl), which is located in a rural region (the largest orchard region in Poland) at a distance of about 60 km in the south-west direction from the city (far from the urban
and industrial developments). The Warsaw measuring site is located in the area, which is a mixture of different surfaces: grass, trees, concrete constructions (buildings, pedestrian footpaths), and asphalt roads.

BS064 is an older generation instrument - Mark II type, which is equipped with the single monochromator. Its spectral range is 290-325 nm in 0.5 nm steps and the spectral resolution 0.6 nm (FWHM). The spectra accuracy decreases for greater values of AOD and for larger solar zenith angles, i.e. for cases with enlarged contribution of the diffuse
component in the total UV radiation, that increases the stray-light effect on the instrument. Furthermore, it does not have a ventilation system. The quality control of its performance has been assessed by almost yearly calibration against the travelling world standard BS No. 17 . BS No. 17 itself is regularly compared with the set of three Brewer instruments, so-





called "Brewer reference triad" (Fioletov et al., 2005). BS No. 64 was also compared with Bentham DM-150 during the project Quality Assurance of Spectral Ultraviolet Measurements (QUASUME) in May 2014 (Gröbner et al., 2005, 2006). The estimated 1σ uncertainty of the erythemal irradiance is about 5%.

BS207 is the newest type - Mark III, that is equipped with a double monochromator reducing significantly the stray-light effect. It is also equipped with the ventilation system, which prevents overheating of the instrument during hotter days. The spectral range is 290-363nm in 0.5 nm steps with almost similar to BS064 spectral resolution. BS207 is calibrated with a set of standard lamps yielding ~5% of measurement spectrum error. For both instruments the SHICRivm software has been used to extend the spectra up to 400 nm and to eliminate erroneous spectra (Slaper et al., 1995).

       The erythemal irradiance is calculated as the integral over wavelength of the 290-400 nm BS spectra after the
SHICRivm standardization, which are weighted by the erythemal action spectrum. The integration for UV-A1 irradiance is without weighting. The erythemal action spectrum follows CIE (1987). Further the partly daily erythemal and UV-A1 doses are calculated as the time integral of the pertaining irradiances for 6h period for all sky-conditions (noon ± 3h) and 3h period for cloudless-sky conditions (noon-3.5h, noon-0.5h). Solar noon is computed from the astronomical formulas corresponding to the specific day of the year. Cloudless-sky conditions are identified using the following two steps algorithm. First step is
the approximate searching for such days using the following criterion: solar UV irradiance derivative with solar zenith angle is negative. In the next step the smoothness of the time series for the day, which meet the first criterion, is examined, i.e. the bell-shape of the UV spectrum must be preserved. In case of jumps in the series such day is omitted.

       Ratios between doses measured by BS064 and BS207, based on collocated observations at Belsk in the period October 2010 – April 2013, allow us to estimate the uncertainty range for the ratio related to differences in BS instrumental
characteristics and in time of observations. The BS measurements are not synchronized as the spectrum length is different. The same ratio is measured in the period of the Warsaw observations (May 2013 to December 2015) by BS207 to assess the urban agglomeration impact on the erythemal and UV-A1 radiation.

       Numerical simulations for the cloudless sky conditions of the ratio between the partly daily doses measured in Warsaw and at Belsk permit us to estimate the UV differences between the sites, caused by the geographical location, as
Belsk is more to the south, $TO_3$, and the surface albedo (lower for surfaces covered by plants). The FastRT model for cloudless sky is used to calculate the spectral irradiance (Engelsen and Kylling, 2005) and erythemal doses. Following model input parameters are selected for such calculations: daily mean total ozone routinely measured by BS, fixed aerosols characteristics representing the mean values derived from the Belsk's CIMEL sunphotometer (AOD at 340 nm=0.32, single scattering albedo=0.92), fixed albedo of 0.03 for Belsk and a set of albedo values for Warsaw that is typical for the various
urban surfaces.





## 3    Results

### 3.1    Instruments comparison at Belsk

In the period from October 2010 to April 2013 both BSs were working simultaneously at Belsk. Figure 1a shows the time series of the measured ratio (BS064/BS207) between the 6h erythemal all-sky doses. The mean value of the ratio is

$1.02 \pm 0.07(1\sigma)$. Figure 1b illustrates that the 1-1 relation between the doses is appropriate for the whole range of the measured irradiances. The coefficient of determination based on this data set is 0.99. The mean ratio for UV-A1 range is $1.01 \pm 0.07(1\sigma)$ for all-sky conditions.

The most of the differences lie within $\pm 5\%$ range but sometimes outliers greater than 10% appear as the measurements were not synchronous. It is difficult to have synchronized measurements by our BSs as scanning time is

different, because of the length of the spectrum, i.e. 290-325 nm for BS064 and 290-363 nm for BS207. BS 064 measures UV spectrum three times per hour, while BS 207 only two times per hour. Thus local cloudiness may be a source of large standard deviations of the mean ratios calculated during-all sky conditions. To remove the effect of cloudiness, we analyze the ratios derived from 3h cloudless-sky measurements before solar noon. The cloudless-sky doses were calculated for the shorter period compared to those for the all-sky conditions as cloudless-sky conditions in Poland usually prevail before noon.

Figure 2a shows the time series of the measured BS064/BS207 ratio for the cloudless-sky conditions and the corresponding scatter plot (Fig.2b). The mean value of the ratio is $1.01 \pm 0.03$ $(1\sigma)$ and there is almost a 1-1 relation between the erythemal doses by both BSs. That is also supported by high value (0.998) of the coefficient of determination. For UV-A1 doses, the ratio is $1.00 \pm 0.04$ $(1\sigma)$. Thus performance of BS064 and BS207 was practically the same during the Belsk's intercomparison. The agreement between output of both BSs was almost perfect, suggesting that the instrumental differences

did not have much influence on the ratio between the doses.

### 3.2    Numerical simulations

Part of the measured difference between both BSs, may be a result of the more northern location of the Warsaw site comparing to the Belsk site, different $TO_3$ over the sites, and surface albedo. In this sub-section, the modelled cloudless-sky doses over 6h period (+/- 3h to solar noon) are analyzed for Warsaw and Belsk. FastRT (Engelsen and Kylling, 2005) is used

in the simulations taking into account real $TO_3$ values (i.e. daily mean $TO_3$ measured by BS using the so-called direct sun observations), fixed aerosol characteristics based on averaged results of CIMEL observations at Belsk, and prescribed values of surface albedo equal to 0.03 at Belsk and a set {0.03, 0.06, 0.12} in Warsaw.

Figure 3a shows the time series of the daily ratio of $TO_3$ measured at Belsk and in Warsaw. Figure 3b illustrates the scatter plot of the daily data. The mean ratio (BS064/BS207) from all data points presented in Fig.3a is $1.00 \pm 0.01$ $(1\sigma)$. The

correspondence between $TO_3$ values is found for the whole range of $TO_3$ variability at Belsk. All data points are in close proximity to the diagonal line representing the 1-1 relationship between the variables shown in the scatter plot. Thus, $TO_3$ is not a factor responsible for the UV difference between the sites. The erythemal doses calculated for the 6h near noon period





with use of these ozone values, give the mean ratio $1.02 \pm 0.02 (1\sigma)$ for simulations assuming the surface albedo is equal to 0.03 (typical for rural regions) for both sites (see Fig.4 for the results with the mark "alb=0.03"). Thus, the $TO_3$ spatial variability and the geographical location of the site contribute only slightly to the UV differences between the sites.

Usually (in snowless periods) the urban albedo is higher than that in rural site and it provides somewhat higher intensity of
the surface UV. Figure 4 presents simulations of the ratio between 6h near noon erythemal doses between the sites assuming the surface albedo is equal to 0.03 at Belsk but 0.06 or 0.12 in Warsaw. We have selected these values for the urban albedo in UV range based on Castro et al. (2001) experimental data over asphalt sites and grey surface-cement sites respectively, taken in Mexico City metropolitan area. The mean ratio derived from all the data points shown in Fig.4 is $1.00 \pm 0.02$ $(1\sigma)$ and $0.98 \pm 0.02$ $(1\sigma)$, for the Warsaw surface albedo equal to 0.06 and 0.12, respectively. The surrounding of the Warsaw
measuring site is a mixture of areas with grass, asphalt, and grey-cement surface, so it could be assumed that surface albedo in the surrounding of the station is of 0.06. It means that a slightly higher albedo over the urban site compensates the latitudinal effect of the UVR differences between the sites.

### 3.3    Comparison of erythemal and UV-A1 doses measured at Belsk and in Warsaw

BSs were working simultaneously in Warsaw and at Belsk in the period from May 2013 to December 2015. The
erythemal and UV-A1 doses calculated for these sites in the periods symmetrical around local noon for 6h for all-sky and 3h before noon for cloudless sky conditions, are analyzed to find the Belsk-Warsaw ratio between the measured doses (BS064/BS207). If the ratio obtained during the cloudless-sky conditions differs significantly from that previously obtained during the cloudless-sky conditions for BSs operating in Belsk, it will allow us to estimate the urban aerosols effect on the surface UVR. The similar approach with the use of all-sky data will provide an estimate of the urban cloud effect on the
surface UVR.

Figure 5 shows the time series of BS064/BS207 measured ratio for the erythemal doses (Fig. 5a) and the UV-A1 (Fig. 5b) doses for cloudless-sky conditions simultaneously appearing both in Warsaw and at Belsk during 3h before local noon. The ratio oscillates around 1 within the range ~0.9-1.1. The main reason for this scatter is the interpolated erythemal (or UV-A1) irradiance value at the beginning (3.5 h before local noon) and at the end (local noon-0.5h) of the calculated
period. BS observations were rarely made exactly at the starting and ending moments. Thus linear interpolated values were used taken from observations closest to the beginning or to the end of the period, i.e. the irradiance values just outside the observing period were taken into account. The mean value of the Belsk-Warsaw ratio is $1.06 \pm 0.04$ $(1\sigma)$ and $1.03 \pm 0.04$ $(1\sigma)$ for the erythemal and UV-A1 dose, respectively. The same values calculated for all 6h near noon doses (Fig.6) during all-sky conditions are $1.08 \pm 0.19$ $(1\sigma)$ and $1.05 \pm 0.17(1\sigma)$, respectively. Much larger uncertainty ranges of the estimates for
all-sky conditions are due to the cloudiness effects but the mean values of the ratio are only slightly larger than those found during the Belsk's intercomparison of the instruments. In spite of possible different cloud properties over Belsk and Warsaw during 6h measurements, the determination coefficient values are still high, i.e. equal to 0.96 and 0.95 for the erythemal and UV-A1 doses. The 1-1 correspondence between doses is kept for the whole range of the data (Fig.7).



Standard statistical test for the difference in the mean values taken from two large samples of unknown distribution (Daniel and Cross, 2013) is used to evaluate, if a higher BS064/BS207 mean ratio during Belsk-Warsaw comparison of BSs relative to the mean ratio found during Belsk intercomparison is statistically significant. The working hypothesis, that the mean values of BS064/BS207 ratio (both for cloudless-sky and all-sky conditions) are higher during the Belsk-Warsaw comparison, is supported by the test at the significance level > 0.01. We conclude that the urban agglomeration only slightly attenuate the erythemal and UV-A1 radiation. The aerosol effects are responsible for about 6% (or 3%) larger 3h erythemal (or UV-A1) doses at Belsk and cloud effects only slightly enlarge the Belsk-Warsaw difference, i.e. ~2% both for the erythemal and UV-A1 doses.

## 4    Discussion and conclusions

Warsaw agglomeration has over 3.5 million population with high pollution due to the heavy vehicle emission and industry (mainly electric powers), causing numerous cases over the EU air quality threshold. Like other large cities, it is expected that Warsaw produces the well known heat island that makes specific boundary layer allowing anthropogenic aerosols to reach higher atmospheric layers that may enhance AOD and affect cloud properties (e.g. level of cloudiness, droplet size, liquid water content). Previous study of the Belsk-Warsaw differences in the aerosols properties (Zawadzka et al., 2010) revealed similar values of the Angström exponent and higher Warsaw AOD values at 500 nm of about 0.02 i.e., ~10% of the overall mean AOD at Belsk for this wavelength. Similar values were reported by Chubarova et al. (2011) analyzing the results of aerosols measurements by CIMEL sunphotometers located in Moscow (megacity with population over 10 million) and Zvenigorod.

Using the value of the erythemal radiation amplification factor due to aerosols of 0.15, which is defined for AOD at 550 nm (Krzyścin and Puchalski, 1998), it could be estimated that 10% AOD difference between Belsk and Warsaw would induce ~2% attenuation of the erythemal doses in Warsaw. The present study, which is based on the measured UV spectra by BSs, shows higher difference (~6%) of the erythemal irradiance under cloudless-sky conditions due to the urban pollution effect. It seems possible that higher absorption of UV irradiance, i.e. smaller SSA by the anthropogenic aerosols, is a factor that may be responsible for larger attenuation of the surface UVR in Warsaw.

Fountoulakis et al. (2016) discussed factors important for the UV spectra variability in Thessaloniki. They pointed out that the cloudless-sky UV-A irradiances could be sensitive not only to AOD changes but also to SSA changes. Chubarova et al. (2011) analyzing results by CIMEL sun photometers located in Moscow and in Zvenigorod (less polluted site) found that uncertainty range of SSA is too high, precluding discussion of the SSA urban effects. However, they found that SSA in Moscow for the visible range of solar radiation was 0.02-0.03 smaller than that obtained over the clean site. It is worth mentioning that there is a lack of the direct retrieval to obtain SSA from spectral measurements. Indirect method for BS has been proposed (Bais et al., 2005) depending on the assumed values of the asymmetry parameter, surface albedo, aerosol vertical profile, and the extraterrestrial solar spectrum. They found that frequently low SSA values < 0.7 were



observed in Thessaloniki when AOD at 340 nm was less than 0.3. It could induce 10%-20% attenuation of the irradiance at 340 nm relative to the case of SSA=0.92 being typical for clean sites. Based on their results we could estimate that ~5% decline of UV irradiance, which is attributed to the more absorbing Warsaw aerosols, is caused by aerosols with SSA ~0.8 at 340 nm. Such estimate looks probable as the Warsaw observing site is among the most polluted parts of the city because of

the abnormal vehicle emission in the nearby main city roads. Kazadzis et al. (2009b) found that UV-A irradiance increase in the Thessaloniki for the period 1998- 2006 cannot be explained only by the AOD changes, but also by the changes of SSA over the area, due to improvement of the air quality there.

For all-sky conditions the attenuation of the surface UV radiation in Warsaw is only slightly higher than that found during cloudless-sky conditions, i.e. ~8% for erythemal doses and 5% for UV-A1 doses, which is ~2% greater than found for the

cloudless-sky case. Cloud effects should be more pronounced during warm period of the year where the city heat island may generate stronger convection than that existing in the cold period. Romanov (1999) analyzing NOAA satellite images retrieved higher cloud cover in summer over the central Moscow compared to its suburbs. Inoue and Kimura (2004) found that in Tokyo there were more low level clouds in the summer period (July-August) comparing to rural sites in Kanto region. Moreover, urban heat islands lead to more thunderstorm initiation episodes (e.g. Sheperd, 2005; Haberlie et al., 2015).

The classical theory (Twomey, 1977) states that when there are more aerosols high above the surface due to stronger updraft generated by the city warm island, aerosols serve as cloud condensation nuclei, reduce the size of cloud effective radius and increase number of droplets causing larger cloud optical thickness (COT) and finally higher attenuation of radiation reaching the Earth's surface. Thus additional cloudiness generated over large cities may act as an umbrella against excessive UV radiation. We calculate BS064/BS207 ratio during Belsk-Warsaw comparison campaign taking into

account measurements in the warm period, 15 May – 15 September. We expect to find a higher ratio for that period according to the classical theory stated above. However the ratio is only slightly lower, i.e. $1.06 \pm 0.17(1\sigma)$, for the erythemal doses, and $1.05 \pm 0.16(1\sigma)$ for UV-A1 doses. It may suggest that contrary to the expectation, COT is smaller over the urban area. Jin et al. (2005) discussed aerosol-cloud relationship over New York and Huston. They found that thick urban aerosols correspond to low COT there. Thus it seems possible that even higher cloudiness over urban areas does not mean higher

attenuation of solar radiation, because the urban aerosols modify the cloud structure compensating the effect of increased cloud cover there.

A slightly higher BSO064/BS207 ratio is found during the cold period because of an effect of seasonal albedo changes, i.e. value $1.09 \pm 0.20(1\sigma)$ was obtain using the winter data comparing to $1.08 \pm 0.19 (1\sigma)$ found in the all year data. During winter, snow cover melts faster in the centre of Warsaw, due to the city warm island. Furthermore, snow is

systematically removed from the roads and pavements in the city. Snow during the Warsaw-Belsk comparison campaign was hardly observed thus the ratio was affected only slightly by the seasonal albedo differences.

Zerefos et al. (2012) discussed that present UV level over many sites in the northern mid-latitudes is high due to the positive trends in the UVR related to the decreasing total ozone (up to mid 1990s) and negative trends in amount of clouds and aerosols (up to mid 2000s). Figure 8 shows the annual erythemal means for the period 1976-2015 based on the updated





(for the period 2009-2015) time series of the broad-band UV measurements at Belsk (Krzyścin et al., 2011). The UV increase in Poland for the period 1976-2015 is especially strong as the cloudiness and amount of aerosols in the atmosphere in the mid 1970 and early 1980s was large, because of the enormous environment pollution during the communist era. The UV level stabilizes around 2006 and the present level of the erythemal irradiance at Belsk is ~15% larger than that in the mid 1970s.

Our study proves that the UV level in Warsaw is only slightly lower (~5%-8%) than that found in cleaner surroundings of Warsaw. It seems that higher absorption of UV radiation by urban aerosols (lower SSA values) is the main source of the urban UVR attenuation there. However, Parisi et al. (2004) found that over some non shaded part of the city with high albedo (e.g. concrete surface) there is an amplification of the human exposure of up to 7% for people in the upright position. Thus contaminated urban atmosphere over Warsaw cannot be treated as a shield against excessive human exposure.

**Acknowledgements**. This work was partially supported within statutory activities No. 3841/E-41/S/2016 of the Ministry of Science and Higher Education of Poland. We appreciate a support from early-carrier scientist grant (A. Czerwińska) No. 500-10-18 by the Institute of Geophysics, Polish Academy of Sciences.

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




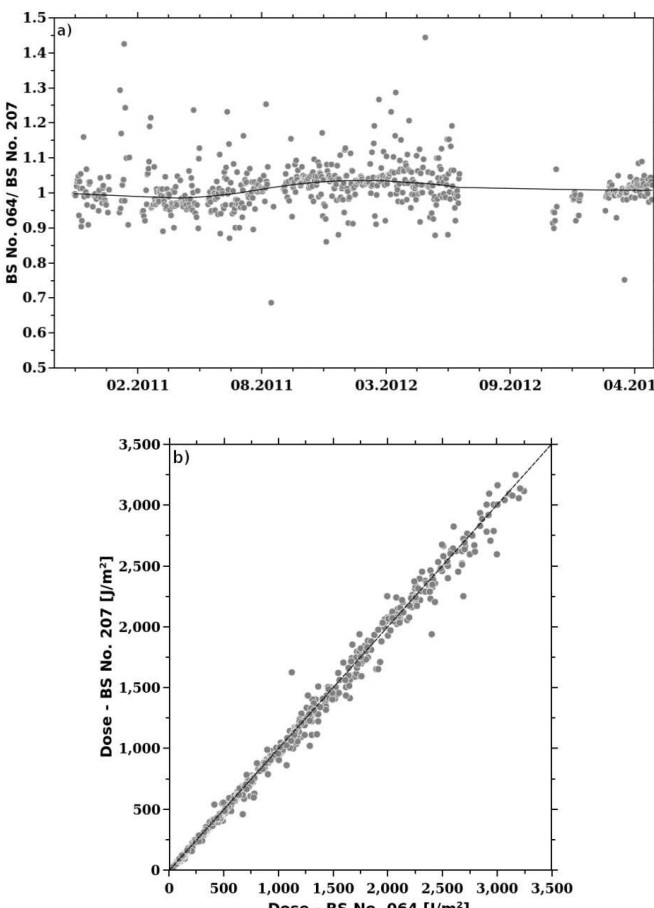

**Figure 1a: The ratio between erythemal 6h (noon+/-3 hr) doses measured by the Brewer Spectrophotometer No.64 and No.207 while working simultaneously at Belsk (all-sky conditions). The solid curve represent the smoothed data by LOWESS filter.**

**Figure 1b: The doses measured by the Brewer spectrophotometer No.207 versus those measured by the Brewer spectrophotometer No.64.**




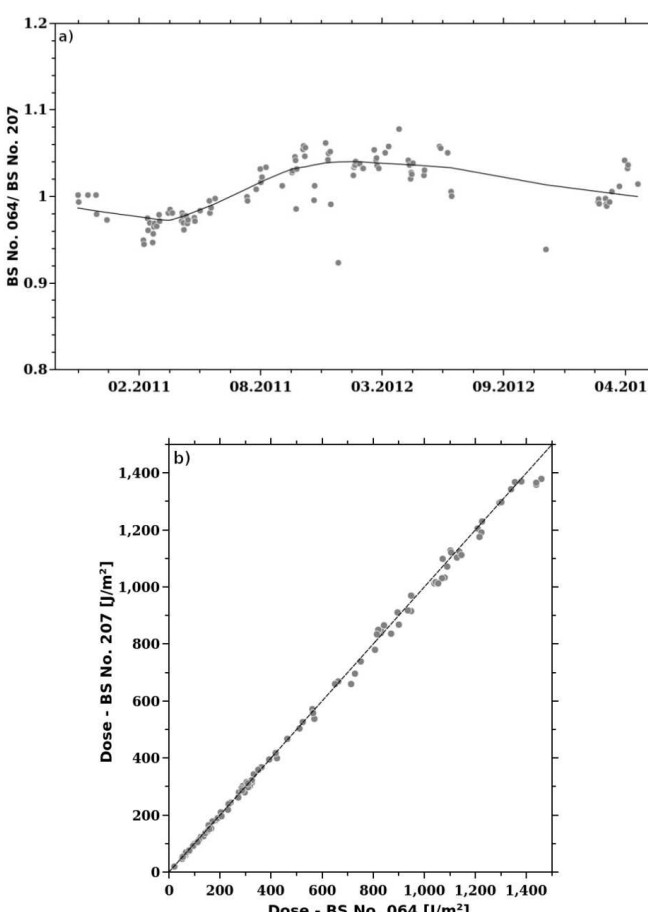

**Figure 2a: The same as Fig.1a but for cloudless-sky conditions and 3h doses calculated for the period before noon (noon-3.5h, noon-0.5h).**

5    **Figure 2b: The same as Fig.1b but for cloudless-sky conditions and 3h doses calculated for the period before noon (noon-3.5h, noon-0.5h).**





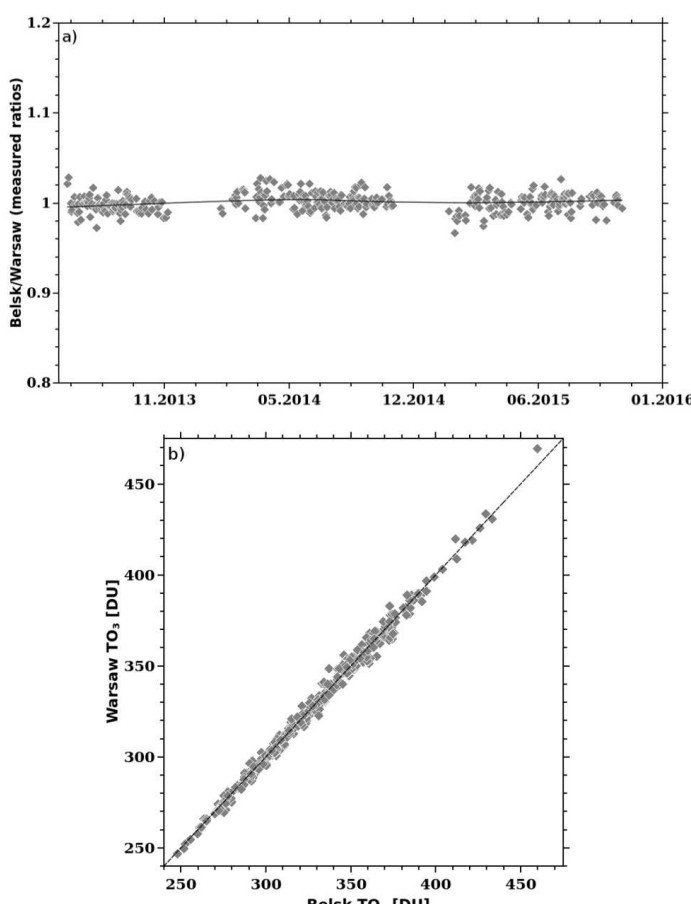

**Figure 3a:** The ratio between total ozone values measured by the Brewer Spectrophotometer No.64 and No.207 while working simultaneously at Belsk and in Warsaw for the period May 2013-December 2015. The solid curve represent the smoothed data by
5   **LOWESS filter.**

**Figure 3b:** Total ozone values measured by the Brewer Spectrophotometer No.64 versus total ozone values measured by the Brewer spectrophotometer No.207 while working simultaneously at Belsk and in Warsaw for the period May 2013-December 2015.





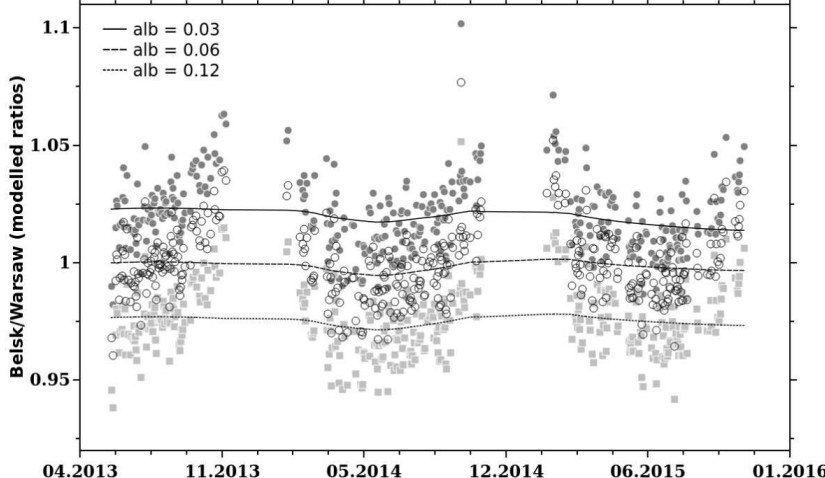

**Figure 4: The Belsk/Warsaw ratio between partially daily 6 hr doses (local noon +/- 3h) calculated by FastRT model with following input values: total column amount of ozone measured by the Brewer spectrophotometer, SSA=0.92, AOD=0.32 at 340 nm, surface albedo 0.03 at Belsk and {0.03, 0.06, 0.12} in Warsaw. The solid curve represent the smoothed data by LOWESS filter.**





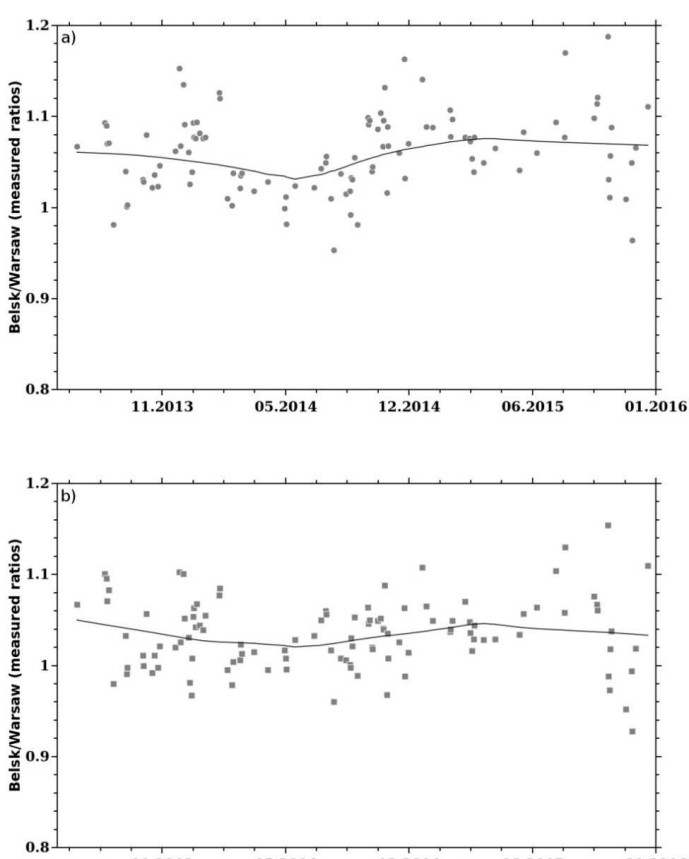

**Figure 5: The Belsk/Warsaw ratio between the partially daily 3h doses (noon-3.5h, noon-0.5h) measured during cloudless-sky conditions existing over both sites for erythemal doses (a) and UV-A1 doses (b). Solid curves represent the smoothed data by LOWESS filter.**





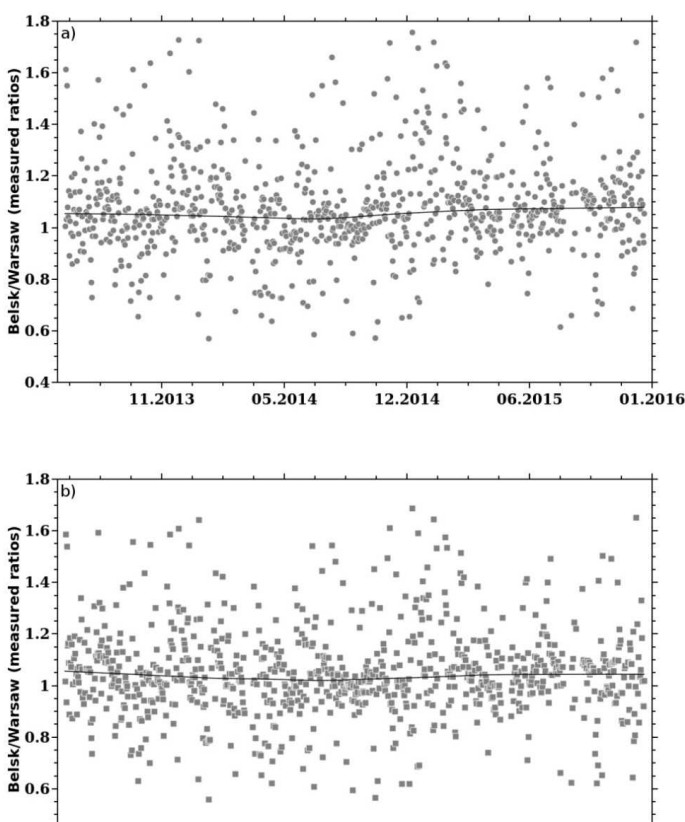

**Figure 6: The same as Fig.5 but for the near noon partially daily dose (noon-3h, noon+3h) for all-sky conditions. Solid curves represent the smoothed data by LOWESS filter.**




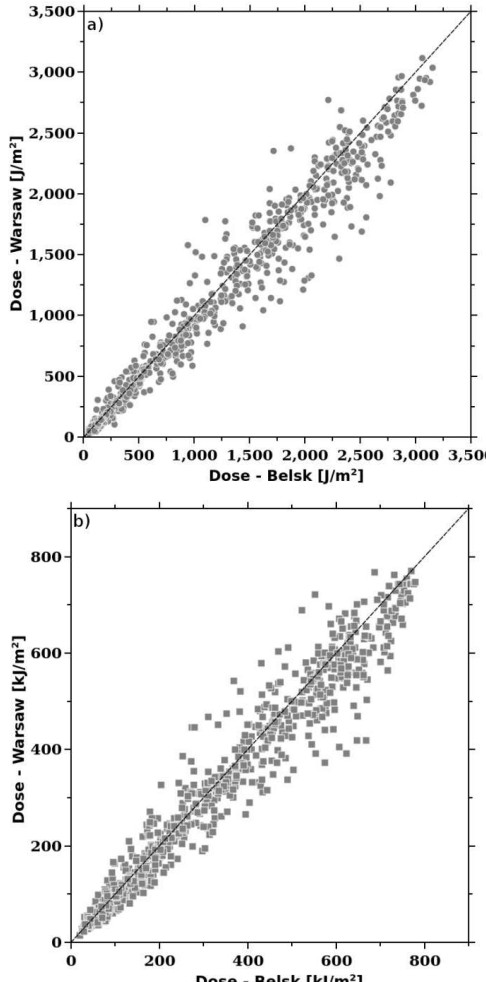

**Figure 7: Partially daily near noon doses {local noon-3h, local noon+3h} measured in Warsaw versus those at Belsk: erythemal doses (a) and UV-A1 doses (b).**





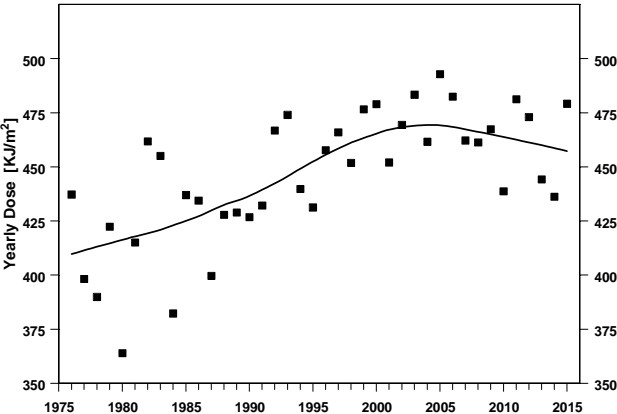

**Figure 8: Yearly means of the erythemal dose measured for the period 1976-2015 at Belsk (squares). The solid curve denote the smoothed data by LOWESS filter.**

