# Peer review of "Effects of urban agglomeration on surface UV doses: a comparison of Brewer measurements in Warsaw and Belsk, Poland, for the period 2013-2015"

_Atmospheric Chemistry and Physics, 2016_

## Editor Comment (EC1) · S. Kazadzis (Editor) · 28 Jun 2016

Here are my comments concerning the publication to ACP

line 9: you have to explain what exactly you mean smoothness. What is the algorithm ?

Figures 1 and 2. How much stray light issues mentioned before in the text affect the ratio especially on low elevation (winter) and cloudy (low signal) conditions ?

Why a 6h cloudless period with the model when you measure 3h with the BS ?

[Figure]

analysis: 1. The real day to day AOD has to be used in order to quantify the real AOD effect 2. A sensitivity study has to be included in order to specify the effect of ozone variability within the 6 hour period to the model results 3. Using the same TOC for the two locations the solar zenith angle effect of the 60 km distance on erythemal dose can be exactly quantified with the model help. 4. Figure 3 TOC differ within 5 % which can be ~15 DU. This can not be considered negligible. 5. There is a clear solar zenith angle dependence on the ratios in the order of 5% (for all albedos) probably related with the solar zenith angle differences 6. A 6 to 12 % albedo in the UV without snow. Is there any publication or theoretical document to support this?

AOD from cimel you have to specify the wavelength and the level (1.5 or 2 ) of data used and also to mention that Cimel SSA is measured at the visible region and you have assumed that it can be extrapolated to the UVB. In addition there is no documentation for the uncertainties of CIMEL SSA for AOD <0.4 so since you are using it you should comment on this.

Figure 5. Erythemal : there is a clear solar zenith angle dependence of the ratio. You can show this if you plot this raio against minimum solar zenith angle for each of the days used.

Discussion

Line 11: It is not straight forward to extrapolated AOD amplification factors and percentages from 550nm to the UV.

The paper needs clear restructuring in order to quantify different effects: Here we have spatial and temporal related differences mixed that are also linked with AOD, ozone, and albedo variability.

First issue is the 60Km distance. Using the model you can quantify this. It is related with the 6h window and also ozone (and partly aerosol effects). To make things easier I would suggest to use a constant solar zenith angle (e.g. X +/- 1 degree) for both places

in the comparisons to get rid of this problem or to try to homogenize the series based on the model results. In addition, using a conctant solar angle you get rid of problems like ozone variability over the 6 hour period, AOD changes, averaging (measurement frequency) issues. What remains is a. the ozone difference, b. the AOD difference, c. the albedo possible differences d. instrumental issues such as stray light and absolute calibration. Its important to try to separate them for example starting from UVA where ozone plays no role so to quantify the AOD effects.

Also working in a constant solar zenith angle provides the possibility to calculate indirectly the AOD that has to be used in order to match the Belsk and Warsaw measurements for a constant SSA. Then to compare your results with the MODIS related study. In the end if all do not add up you can quantify the SSA needed to be used in order to match the measurements of the two sites.

SHICRIVM: Since you are not actually measuring the UVA1 but you are using SHICRIVM to simulate the spectrum, this adds an additional uncertainty especially for the single monochromator measurements.

---

## Referee Comment (RC1) · Anonymous Referee #1 · 10 Jul 2016

**General comments**

The paper by Czerwińska et al provides useful information regarding the effect of an urban agglomeration on the levels of the solar UV irradiance. It highlights the importance of the aerosol optical properties for the determination of the UV irradiance that reaches the earth surface and has the potential to contribute in the better understanding of the complex interactions between aerosols, clouds, surface albedo and UV radiation in an urban environment. The authors compare the erythemal and UV-A1 (340-400 nm) doses measured by the Brewer spectrophotometers in Warsaw (52.3°N, 21.0°E) and Belsk (51.8°N, 20.8°E) and are trying to quantify the effects of differences in surface albedo, cloudiness, aerosol optical depth and aerosol single scattering albedo.

However, the main problem in the data analysis is that the effect of different latitude (thus of different SZAs for the measured UV doses) has not been removed (or quantified) properly leading to biases in the quantification of the effect of other factors (such as the aerosol SSA and the cloudiness). The effect of different latitude changes periodically in the year and is more pronounced in winter (higher SZAs, which means that the difference of 0.5° becomes more important). The authors considered the effect of different SZAs to be invariant during the year which is not correct. Thus, I suggest to re-analyze the data and either quantify the effect of different latitude properly (e.g. with the use of a radiative transfer model) or perform the comparison for standard SZAs.

The second important issue that has to be solved prior to publication in ACP is the large number of editorial, grammatical and linguistic errors in the manuscript. The authors have to try hard to improve the manuscript. If possible, I suggest that the manuscript should be edited by a native English speaker.

Additionally, I suggest reorganizing sections 3 and 4 as follows: 3.1. Comparison between measurements at Belsk, 3.2. Comparison between measurements at Belsk and Warsaw, 3.3. Quantification of the factors which are responsible for the differences, 3.4. Long-term change of the erythemal irradiance at Belsk. In section 3.3 you can include the numerical simulations (now section 3.3) and part of the discussion from section 4 (e.g. the results reported in P8, L15 – 31 regarding the effects of cloudiness). I also suggest moving figure 8 (and the relative discussion) in section 3.4 and expand the discussion. This way, I believe that it will be easier for the reader to follow the discussion.

The effect of different SZA (presented in paragraph 3.2) should be initially studied for UV-A1. Then, the combined effect of SZA and TO3 could be studied for the erythemal dose. Although the effect of SZA is stronger for lower wavelengths (due to stronger Rayleigh scattering and increased absorption by TO3 for larger SZAs) and the effects of SZA and TO3 on erythemal irradiance are not independent to each other, this way you could get a quantitative estimation regarding the effect of differences in TO3. However, the most effective way to quantify the effect of different TO3 is to study the ratios of UV-A1 and erythemal irradiance for specific SZAs (or small SZA intervals).

**Specific comments**

Please define a specific acronym to use in the document for each Brewer. For example, Brewer with Serial Number 207 is referred as BS No. 207, BS 207 and BS207 (without defining what the number 207

is) at different points of the document. Please choose a single acronym and use it everywhere (in the manuscript and the figures). E.g. you could define at the beginning of the methodology section that each BS with serial number xxx will be referred as BSxxx and then refer to each Brewer the same way.

**Abstract**

P1, L7-8: replace "well-know" with "well-known"

P1, L8: replace "cleaner" with "less polluted"

P1, L9-11: replace the sentence:

"The present study focuses on differences in the erythemal and UV-A1 (340-400 nm) doses measured by the Brewer spectrophotometers in Warsaw (52.3°N, 21.0°E) and at Belsk (51.8°N, 20.8°E), which is located in a rural region at a distance of about 60 km in the south-west direction from the city."

With

"The present study focuses on differences between the erythemal and UV-A1 (340-400 nm) doses measured by the Brewer spectrophotometers in Warsaw (52.3°N, 21.0°E) and Belsk (51.8°N, 20.8°E). The latter is a rural region at a distance of about 60 km south-west of the city of Warsaw."

P1, L18: replace "by larger aerosol absorption" with "mainly by larger aerosol absorption over Warsaw"

P1, L18-19: The meaning of the phrase: "It appears that a slightly increased optical depth of the urban aerosols and properties of clouds generated over Warsaw are less important for the UV attenuation." is not clear at this point. I would suggest replacing with something like:

"Differences between the aerosol optical depth and cloud optical properties over the two sites are found to be less important."

P1, L19-20: replace "In this work we are showing that the higher city surface albedo compensates for the solar UV attenuation caused by urban aerosol load in the city of Warsaw."

With something like:

"We show that the higher surface albedo in Warsaw compensates for the stronger attenuation of the solar UV radiation by the urban aerosols."

**Introduction**

P2, L5: add appropriate references

P2, L6: add appropriate references

P2, L8: replace "depended" with "dependent"

P2, L11: replace "surface UV attenuation" with "attenuation of the solar UV radiation"

P2, L10-12: add references to support your statement that "The absorption by SO2 (in the UV-B range) and NO2 (mostly in the UV-A range) is important for the surface UV attenuation only in extreme concentrations of such gases."

P2, L12: Replace "surface intensity of UV" with "intensity of the solar UV radiation at the earth surface". Furthermore, in addition to the properties, the amount of aerosols and clouds also affect UV radiation.

P2, L12-14: Change the phrase: "The negative trends in these variables, found over many of the northern hemisphere mid-latitudinal sites in the 1989s and 1990s, lead to increases of both the UV-B and UV-A irradiance"

With

"Increases of both the UV-B and UV-A irradiance have been reported over several mid-latitudinal sites of the northern hemisphere since the beginning of the 1990s, which have been mainly attributed to decreasing attenuation by aerosols and clouds."

P2, L15: Replace "An" with "The".

P2, L15: Replace "the large urban agglomeration" with "large urban agglomerations".

P2, L17: Replace "UV cloudless sky irradiances" with cloudless-sky UV irradiances". Also replace "its suburbs" with "a sub-urban area near Athens".

P2, L19: Replace "The erythemal irradiance at the centre of Athens was 30% lower than at the suburbs with similar values of total ozone (TO3) for days with increased pollution in the air."

With

"The erythemal irradiance at the centre of Athens was up to 30% lower than at the outskirt site during days with increased air pollution over Athens basin and similar values of total ozone (TO3) over the two sites."

P2, L20-21: What you write here is not clear. Please be more specific. Do you mean differences from the measurements or differences by corresponding simulations over the Athens basin?

P2, L23: Delete "the" before "winter" and "summer". Replace "Mexico" with "Mexico City".

P2, L23-24: Are 9% and 21% the differences between the annual mean levels for winter and summer? Please be more precise.

P2, L24-25: " Corr et al. (2009) … 0.7 – 0.85". This sentence is not clear. Please rephrase.

P2, L26: Delete "atmospheric"

P2, L30: Delete "of aerosols"

P3, L3-8: Notice that for a typical Angström parameter of ~1.5, the differences in AOD becomes ~2 times larger for UV-B wavelengths. I suggest that you should provide quantitative estimations of the changes in UV irradiance due to the reported differences in AOD (e.g. for a low SSA = 0.85) at this point, to prove that the reported differences in AOD do not induce large differences in UV irradiance. That can be easily achieved by performing modeling simulations. Furthermore, it should be mentioned that for organic particles, the absorption in the UV range may be even larger than that predicted by interpolating using the Angström parameter for the visible range of the spectrum (e.g. see Bais et al. (2015)* and references therein).

* Bais, A. F., R. L. McKenzie, G. Bernhard, P. J. Aucamp, M. Ilyas, S. Madronich, and K. Tourpali (2015), Ozone depletion and climate change: impacts on UV radiation, *Photochemical & Photobiological Sciences*, *14*(1), 19-52, doi:10.1039/c4pp90032d.

P3, L4: Remove "," after "stated".

P3, L6: "the difference" instead of "it"

P3, L11: Delete "a specific"

P3, L14: Delete "at"

**Methodology**

 P3,L23: Replace "at Belsk (51.8°N, 20.8°E, 190 m amsl), which is located in a rural region" with "Belsk (51.8°N, 20.8°E, 190 m amsl). The latter is a rural region"

P3, L25: Replace "the area" with "an area"

P3, L28: Replace "spectra accuracy" with "spectral accuracy"

P4, L3: Please provide reference(s) regarding the fact that the estimated uncertainty in the erythemal irradiance is 5%.

P4, L8: In this section you describe how each Brewer (207 and 64) is calibrated. Was the calibration procedure for the two Brewers the same before and after BS207 was moved to Warsaw? Is BS207 also calibrated against BS017? Please add some more information to convince the reader that the changes in the ratio are not due to a change in the calibration procedure or due to change of the BS207 characteristics during transportation from the one site to the other.

P4, L11: Replace "The erythemal action spectrum follows CIE (1987)" with "The erythemal action spectrum is that suggested by the Commission internationale de l'éclairage (CIE) (CIE, 1987)"

P4, L13: Were there any criteria for the selection of the partly daily doses used for the comparison between Belsk and Warsaw? Is there a minimum amount of measurements (or measurements per 1 or 2 hours) below which the data are rejected? Calculation of the integral from only a small number of measurements and/or large gaps in the 3- or 6-hour period may lead to large differences between the

integrals for the two sites. If not already done, I would suggest using a filter (e.g. use only time intervals with at least one measurement per 1 - 2 hours).

P4, L15: How confident are you for your cloud detection method? Are there cloudy cases that cannot be detected? Can you estimate if, and in what extent, they affect your results?

P4, L23-30: The specific paragraph is carelessly written. Please try to re-write it more carefully.

**Results**

Figure 1: Add some more information in the manuscript for LOWESS filter and/or proper references.

P5, L8-9: I think that the phrase "The most of the differences lie within ±5% range" is not necessary here since in the previous paragraph the 1σ uncertainty (which by the way is 7%) is given.

Figure 2a: In the specific figure there are two data points near 0.9. Is there any explanation for this large difference between the results from the two instruments for the particular days?

P5, L22: Please specify that the higher latitude of the site at Warsaw means that for the same time, the solar zenith angle over Warsaw is always lower by ~0.5° compared to Belsk.

P5, L23: replace "surface albedo" with "different surface albedo"

P5, L25: replace "in" with "to perform"

P5, L27: Replace the phrase "prescribed values of surface albedo equal to 0.03 at Belsk and a set {0.03, 0.06, 0.12} in Warsaw" with "standard values of surface albedo equal to 0.03 for Belsk and 0.03, 0.06 and 0.12 for Warsaw"

P5, L28: Replace "of" before $TO_3$ with "between" and delete "in"

P5, L30: Do you mean "coincidence" instead of "correspondence"? In this case it is for the entire range of the $TO_3$ variability of both sites and not only of Belsk.

P6, L1: "assuming that" instead of "assuming"

Figure 4: There is an obvious annual cycle of the ratio. I suppose that this is due to the stronger effect of the 0.5° difference in SZAs in winter, when the SZAs are larger. Thus, the effect of larger albedo compensates for the effect of different latitude only for a specific period of the year. The same annual cycle is obvious in Figure 5, possibly due to the remaining effect of the difference in SZA. Given that in Figure 4 there are no results for December and January, when the effect of SZA is expected to be even larger, while in Figure 5 there are results for the particular months, I believe that the deviation of the mean ratio from unity is partially due to the effect of different SZAs. I suggest that you should either quantify the remaining effect due to different SZAs and take it into account in the discussion of the results presented in Figures 5 and 6, or alternatively compare the irradiances for specific SZAs (thus slightly different time).

P6, L11: remove "of"

P6, L11: remove "in"

P6, L15: replace

"in the periods symmetrical around local noon for 6h for all-sky and 3h before noon for cloudless sky conditions"

with

"for 6h symmetrical periods around local noon for all-skies, and 3h periods before local noon for cloudless skies"

P6, L17: Delete "previously"

P6,L19: Replace "The" with "A"

P6,L23: It seems to me that the ratio oscillates around ~1.05 and not 1. As already commented, I believe that part of the spread in the calculated ratios is due to the remaining effect of the difference in the latitude of the two sites.

Figure 5: I suppose that the slightly different pattern of the temporal evolution of the ratios for the erythemal doses and the UV-A1 doses are again because of the effect of the different SZA. The effect of different SZAs is stronger for lower wavelengths, being partially responsible for the larger ratios of the erythemal doses compared to those of the UV-A1 doses.

P7, L6: "attenuates" instead of "attenuate"

P7, L10: "emissions" instead of "emission"

P7, L11: add references to support your statement that "causing numerous cases over the EU air quality threshold"

P7, L12: what do you mean with the phrase "makes specific boundary layer"? Please explain (e.g. is the upper limit of the boundary layer higher compared to the nearby rural areas?).

P7, L12-13: Add references

P7, L15: "higher AOD at 500 nm over Warsaw" instead of "higher Warsaw AOD values at 500 nm"

P7, L16: "similar differences" instead of "similar values"

P7, L21:"~2% more attenuation" is more accurate than " ~2% attenuation".

P7, L21-24: Again, these numbers may change after re-evaluation of the effect of the SZA.

P7, L30-31: "An Indirect method for BS has been proposed by Bais et al. (2005)" instead of "Indirect method for BS has been proposed (Bais et al., 2005)"

P8, L2: add appropriate reference to support that the typical SSA for rural sites is 0.92.

P8, L2-4: Since the overall difference is 6% and AOD difference is responsible for 2% (according to what is written in the previous paragraph) then SSA differences should compensate for 4% - and not 5% - of the difference. However, this might be different if you take into account the effect of the SZA.

P8, L5-7: I suggest moving the sentence "Kazadzis et al. (2009b) … quality there" to P7, L27, after "… to SSA changes."

P8, L6: "in Thessaloniki" instead of "in the Thessaloniki"

P8, L20 – 31: As already commented, the effect of difference in SZA is more pronounced during the cold period (larger SZAs) and less pronounced during the warm period (smaller SZAs). The effect of different SZAs has to be removed – or taken into account - properly, so that you can get more accurate conclusions.

Figure 8: I recommend adding a paragraph (e.g. 3.4) and expand the discussion relative to Figure 8. Furthermore, the discussion regarding the agreement of your results with the results of other recent studies (e.g. Zerefos et al. (2012), de Bock et al. (2014), Fountoulakis et al. (2016)), as well as the reasons for this agreement could be expanded.

P8, L24 - 26: "Thus, it seems possible that increased cloudiness over urban areas does not necessarily mean increased attenuation of solar radiation, since modification of the cloud structure and properties by the urban aerosols may lead to the formation of clouds which attenuate the solar radiation less effectively"

 instead of

"Thus it seems possible that even higher cloudiness over urban areas does not mean higher attenuation of solar radiation, because the urban aerosols modify the cloud structure compensating the effect of increased cloud cover there"

P8, L27: "the" instead of "an"

P8, L32: "level" instead of "levels" and "higher than in the past" instead of "high"

P8, L33: Zerefos et al. (2012) discuss the trends of the UVR after the mid-1990s. Thus, in the particular study they do not discuss the increase of the UVR due to the decrease of ozone until the mid-1990s.

P9, L2-3: Add proper reference to support your statement that that the environmental pollution was enormous in the mid-1970 and early-1980. Furthermore, the UV increases because aerosol and clouds decrease relative to the past – not because they were high in the past.

P9, L6: I do not think that 5-8% is "slightly" lower. I suggest removing the phrase "only slightly".

P9, L8: "parts" instead of "part"

---

## Referee Comment (RC2) · U. Feister (Referee) · 29 Aug 2016

**General remarks**

The authors compare broad-band solar UV radiation exposure data derived from measurements by Brewer spectrometers taken in the city of Warsaw and outside the city. The paper is logically separated into sections. The abstract gives an overview of the paper. It mentions its most significant results. The SI is used throughout the paper. The title uses 'UV exposure', though 'UV dose' is used in the text. The same name should be used in the title and in the text. The number of Figures is adequate to illustrate the text. However, as will be addressed in more detail below, Figure captions need to be clear and provide sufficient details on what is presented. If a Figure consists of two graphs, the Figure caption should address both parts separately, for example by numbering them as a) and b). The axis names should be completed to mention the parameters in the graph. It is recommended to apply corrections to English grammar and language style of the manuscript, and also correct for the many writing errors. The method of separation between the albedo effect on the one hand, and the aerosol and cloud effects to UV exposure at the sites on the other hand is not clear. The authors assume that the albedo is 3% at Belsk and 6% at Warsaw. This difference of surface albedo is shown by model calculations to compensate for the difference in solar height (or latitude) between the sites, if the aerosol load would be the same. The remaining differences in the UV measurements are then interpreted as a result of differences in aerosol loads and cloudiness between the sites. Have you checked the real albedo at the sites? How about seasonal differences in albedo for example due to snow cover? The paper is recommended for publication after revision.

Detailed comments

Page 1, line 20: 'increase of UV exposure for peoples' replace by 'higher UV exposure for people'

Page 3, Section 2: Coordinates of the sites and the types of the immediate surroundings should be included here, not only mentioned in the abstract.

Page 3, line 8: 'Its'

Page 3, line 10: replace 'diffusive' by 'diffuse'

Page 3, second paragraph: Is the higher contribution of diffuse irradiance to global irradiance the cause of the higher internal stray light of instruments, or is it just the lower global irradiance?

Page 3, fourth paragraph: Do you really mean 'clear sky conditions' that refer to no

aerosol or low aerosol load, or do you mean 'cloudless conditions'? If you refer to the latter, the changed wording needs to be applied to the whole text.

Conditions of cloudless sky (or clear sky) were separated from the relative increase of irradiance over time around noontime. Using only this criterion, the separated 'cloudless cases' may still contain cases of clouds that do not occlude the sun. Have you checked by cloud observations or cloud imaging data, how good your selection criterion to find real cloudless cases is?

Page 3, line 28: You refer to the erythemal action spectrum by CIE (1987). Probably, you have taken into account the corrections, as discussed by Webb et al. (2011), Photochem. Photobiol. 97, 483 – 486. If so, the citation should be added.

Page 4, line 17: replace 'moment' by 'time period'

Page 5, last paragraph: You state that the main cause of scatter in the interpolated UV irradiance values is the first and last spectrum of the time period. Why did you not leave those two spectra?

Page 6, line 11: replace 'an increase of BS064/BS207' by 'a higher BS064/BS207'

Page 6, line 31: replace 'decline' by 'difference'

Page 7, line 7: 'radiation'

Figure captions are incomplete and partly confusing. The caption of Fig. 3 says 'The same as Fig. 1', but the ratios are calculated for total ozone values measured simultaneously at Belsk and Warsaw'. Does the upper part of Fig. 3 show ratios of erythemal exposure? Is it measured or modelled? Or, does it show ratios between measured column ozone at the sites? The vertical axis only states 'Belsk/Warsaw'. The lower panel does obviously show total ozone at the sites, but it is not mentioned in the Figure caption. Caption of Fig. 4 should mention that it refers to 'modelled ratios'. Figure 5 should mention that it refers to 'measured ratios'.

---

## Author Comment (AC1) · 7 Oct 2016

**Here are my comments concerning the publication to ACP**

**line 9: you have to explain what exactly you mean smoothness. What is the algorithm?**

Answer:

An explanation of the algorithm is contained in the current version of the manuscript, P4, L20-24:
"Cloudless-sky conditions are identified using a two step algorithm. The first step is a preliminary search for such days using the criterion: the solar UV irradiance derivative with solar zenith angle is negative. In the next step, the smoothness of the time series for the day which fulfilled the first criterion, is examined, i.e. the bell-shape of the UV time series must be identified. There is no strict mathematical criterion applied here, but rather an intuitive inspection of the time series shape.".

**Figures 1 and 2. How much stray light issues mentioned before in the text affect the ratio especially on low elevation (winter) and cloudy (low signal) conditions?**

Answer:

We take into consideration only doses around local noon, when the effect of stray light on instruments is the weakest, thus we assume that it does not have a significant influence on the ratio.

**Why a 6h cloudless period with the model when you measure 3h with the BS?**

Answer:

In the revised manuscript, this discrepancy was removed. We did not simulate the ratio between doses, but the ratio between irradiances for a fixed SZA or time. Please, see the section 3.3.

**analysis: 1. The real day to day AOD has to be used in order to quantify the real AOD effect 2. A sensitivity study has to be included in order to specify the effect of ozone variability within the 6 hour period to the model results 3. Using the same TOC for the two locations the solar zenith angle effect of the 60 km distance on erythemal dose can be exactly quantified with the model help. 4. Figure 3 TOC differ within 5 % which can be _15 DU. This can not be considered negligible. 5. There is a clear solar zenith angle dependence on the ratios in the order of 5% (for all albedos) probably related with the solar zenith angle differences 6. A 6 to 12 % albedo in the UV without snow.**
**Is there any publication or theoretical document to support this?**

Answer:

The revised manuscript contains a new section 3.3, where these problems were explained taking into account the Editor's suggestions. The albedo values are based on the reports of Castro et al. (2001). The reference is used in the text (P8, L11): Castro, T., Mar, B., Longoria, R., Ruiz-Suárez, L. G., and Morales, L.: Surface albedo measurements in Mexico City metropolitan area. Atmósfera, 14(2), 69-74, http://www.scielo.org.mx/scielo.php?script=sci_arttext&pid=S0187-62362001000200002&lng=es&tlng=en, 2001.

**AOD from cimel you have to specify the wavelength and the level (1.5 or 2 ) of data used and also to mention that Cimel SSA is measured at the visible region and you have assumed that it can be extrapolated to the UVB. In addition there is no documentation for the uncertainties of CIMEL SSA for AOD <0.4 so since you are using it you should comment on this.**

Answer:

In the revised manuscript, we do not use AOD from CIMEL, but AOD from measurements with MODIS at 550 nm. Measured values of AOD were used in simulations with LibRadtran, where the type of aerosol was selected to rural. As for SSA, we included this information in the text, P5, L6-12:

"Other input parameters are constants representing typical values used in the UV modelling, e.g. albedo of 0.03 for rural surfaces and SSA=0.92, which is a mean value measured by the CIMEL sunphotometer at Belsk (level 1.5 from AERONET – Aerosol Robotic Network) at 440 nm (http://aeronet.gsfc.nasa.gov). We used SSA at 440 nm as a constant for the whole ultraviolet spectrum, as it was found that monthly averages estimated from BS at Uccle were in close agreement with the CIMEL measurements at 440 nm, especially for 320 nm (Nikitidou et al., 2013). Furthermore, Liu et al. (1991) performed Mie calculations for the rural aerosol model (Shettle and Fenn, 1979) and suggested that for this type of aerosol, SSA is approximately independent of wavelength. There are no measurements performed for SSA at the UV wavelength range."

**Figure 5. Erythemal : there is a clear solar zenith angle dependence of the ratio. You can show this if you plot this raio against minimum solar zenith angle for each of the days used.**

Answer:

This issue is discussed in the modelling section (section 3.3 of the current version of the manuscript). Also, we added Figure 6 to show the dependence.

**Discussion**
**Line 11: It is not straight forward to extrapolated AOD amplification factors and percentages from 550nm to the UV.**

Answer:

In this version of the manuscript, the AOD effect on ratios was calculated directly using measured AOD at 550 nm (from MODIS) and LibRadtran, where the type of aerosol was selected to rural.

**The paper needs clear restructuring in order to quantify different effects: Here we have spatial and temporal related differences mixed that are also linked with AOD, ozone, and albedo variability.**
**First issue is the 60Km distance. Using the model you can quantify this. It is related with the 6h window and also ozone (and partly aerosol effects). To make things easier I would suggest to use a constant solar zenith angle (e.g. X +/- 1 degree) for both places in the comparisons to get rid of this problem or to try to homogenize the series based on the model results. In addition, using a conctant solar angle you get rid of problems like ozone variability over the 6 hour period, AOD changes, averaging (measurement frequency) issues. What remains is a. the ozone difference, b. the AOD difference, c. the albedo possible differences d. instrumental issues such as stray light and absolute**
**calibration. Its important to try to separate them for example starting from UVA where ozone plays no role so to quantify the AOD effects.**
**Also working in a constant solar zenith angle provides the possibility to calculate indirectly the AOD that has to be used in order to match the Belsk and Warsaw measurements for a constant SSA. Then to compare your results with the MODIS related study.**
**In the end if all do not add up you can quantify the SSA needed to be used in order to match the measurements of the two sites.**

Answer:

We reorganised the paper and calculated separately all possible factors that may have an impact on the ratios between the sites. The results are in section 3.3 and in the discussion.

**SHICRIVM: Since you are not actually measuring the UVA1 but you are using SHICRIVM to simulate the spectrum, this adds an additional uncertainty especially for the single monochromator measurements.**

Answer:

In the revised paper, we have used a single wavelength (324nm) for UV-A, which is measured directly by both BSs, instead of UV-A1, to eliminate additional uncertainty connected with the SHICRIVM method.

---

## Author Comment (AC2) · 7 Oct 2016

**General comments**

The paper by Czerwińska et al provides useful information regarding the effect of an urban agglomeration on the levels of the solar UV irradiance. It highlights the importance of the aerosol optical properties for the determination of the UV irradiance that reaches the earth surface and has the potential to contribute in the better understanding of the complex interactions between aerosols, clouds, surface albedo and UV radiation in an urban environment. The authors compare the erythemal and UV-A1 (340-400 nm) doses measured by the Brewer spectrophotometers in Warsaw (52.3°N, 21.0°E) and Belsk (51.8°N, 20.8°E) and are trying to quantify the effects of differences in surface albedo, cloudiness, aerosol optical depth and aerosol single scattering albedo.

However, the main problem in the data analysis is that the effect of different latitude (thus of different SZAs for the measured UV doses) has not been removed (or quantified) properly leading to biases in the quantification of the effect of other factors (such as the aerosol SSA and the cloudiness). The effect of different latitude changes periodically in the year and is more pronounced in winter (higher SZAs, which means that the difference of 0.5° becomes more important). The authors considered the effect of different SZAs to be invariant during the year which is not correct. Thus, I suggest to re-analyze the data and either quantify the effect of different latitude properly (e.g. with the use of a radiative transfer model) or perform the comparison for standard SZAs.

Answer:

We revised the modelling part. In the revised manuscript, we took into account the effect of different latitude and re-analysed data with the use of the LibRatran simulations. Please see P7, L6-11:
"The difference in the geographical coordinates for the sites, which are based on the simulations of the erythemal and UV-A irradiances at 10:40 a.m. (i.e. near local noon) throughout 2015 leads to slightly higher values at Belsk. The modelled ratio changes with SZA (Fig. 6). The average ratio over the whole year is $1.03 \pm 0.02$ (1σ) for the erythemal irradiance and $1.02 \pm 0.01$ (1σ) for UV-A (324 nm). For the warm period (from 15 May to 14 September) modelled ratios were $1.01 \pm 0.003$ (1σ) and $1.01 \pm 0.002$ (1σ), but for the cold period (from 15 September to 14 May) modelled ratios were $1.04 \pm 0.01$ (1σ) and $1.03 \pm 0.01$ (1σ) – for erythemal and UV-A (324 nm) irradiances, respectively.".

**The second important issue that has to be solved prior to publication in ACP is the large number of editorial, grammatical and linguistic errors in the manuscript. The authors have to try hard to improve the manuscript. If possible, I suggest that the manuscript should be edited by a native English speaker.**

Answer:

We did our best to improve the manuscript.

**Additionally, I suggest reorganizing sections 3 and 4 as follows: 3.1. Comparison between measurements at Belsk, 3.2. Comparison between measurements at Belsk and Warsaw, 3.3. Quantification of the factors which are responsible for the differences, 3.4. Long-term change of the erythemal irradiance at Belsk. In section 3.3 you can include the numerical simulations (now section 3.3) and part of the discussion from section 4 (e.g. the results reported in P8, L15 – 31 regarding the effects of cloudiness). I also suggest moving figure 8 (and the relative discussion) in section 3.4 and expand the discussion. This way, I believe that it will be easier for the reader to follow the discussion.**

Answer:

The paper was reorganised following the Referee's #1 suggestions.

**The effect of different SZA (presented in paragraph 3.2) should be initially studied for UV-A1. Then, the combined effect of SZA and TO3 could be studied for the erythemal dose. Although the effect of SZA is stronger for lower wavelengths (due to stronger Rayleigh scattering and increased absorption by TO3 for larger SZAs) and the effects of SZA and TO3 on erythemal irradiance are not independent to each other, this way you could get a quantitative estimation regarding the effect of differences in TO3. However, the most effective way to quantify the effect of different TO3 is to study the ratios of UV-A1 and erythemal irradiance for specific SZAs (or small SZA intervals).**

Answer:

We did the calculation for specific SZAs or time. Please see the results in section 3.3.

**Specific comments**

**Please define a specific acronym to use in the document for each Brewer. For example, Brewer with Serial Number 207 is referred as BS No. 207, BS 207 and BS207 (without defining what the number 207 is) at different points of the document. Please choose a single acronym and use it everywhere (in the manuscript and the figures). E.g. you could define at the beginning of the methodology section that each BS with serial number xxx will be referred as BSxxx and then refer to each Brewer the same way.**

Answer:

We changed the acronyms throughout the paper. They were defined on P3, L20-21:
"(...) by the single monochromator BS, serial number 64 (BS064), and in Warsaw since 2013 by the double monochromator BS, serial number 207 (BS207)".

**Abstract**
**P1, L7-8: replace "well-know" with "well-known"**
**P1, L8: replace "cleaner" with "less polluted"**
**P1, L9-11: replace the sentence:**
**"The present study focuses on differences in the erythemal and UV-A1 (340-400 nm) doses measured by the Brewer spectrophotometers in Warsaw (52.3°N, 21.0°E) and at Belsk (51.8°N, 20.8°E), which is located in a rural region at a distance of about 60 km in the south-west direction from the city."**
**With**
**"The present study focuses on differences between the erythemal and UV-A1 (340-400 nm) doses measured by the Brewer spectrophotometers in Warsaw (52.3°N, 21.0°E) and Belsk (51.8°N, 20.8°E). The latter is a rural region at a distance of about 60 km south-west of the city of Warsaw."**
**P1, L18: replace "by larger aerosol absorption" with "mainly by larger aerosol absorption over Warsaw"**
**P1, L18-19: The meaning of the phrase: "It appears that a slightly increased optical depth of the urban aerosols and properties of clouds generated over Warsaw are less important for the UV attenuation." is not clear at this point. I would suggest replacing with something like:**
**"Differences between the aerosol optical depth and cloud optical properties over the two sites are found to be less important."**
**P1, L19-20: replace "In this work we are showing that the higher city surface albedo compensates for the solar UV attenuation caused by urban aerosol load in the city of Warsaw."**
**With something like:**
**"We show that the higher surface albedo in Warsaw compensates for the stronger attenuation of the solar UV radiation by the urban aerosols."**

Answer:

Suggested changes were made.

**Introduction**
**P2, L5: add appropriate references**
**P2, L6: add appropriate references**

Answer:

The references were added on P2, L9:
Greinert, R., de Vries, E., Erdmann, F., Espina, C., Auvinen, A., Kesminiene, A., and Schuz, J.: European Code against Cancer 4th edition: Ultraviolet radiation and cancer, Cancer Epidemiol. Biomarkers Prev., 39, S75-S83, 2015,
Marionnet C., Pierrard, C., Golebiewski, C., Bernerd, F. et al.: Diversity of Biological Effects Induced by Longwave UVA Rays (UVA1) in Reconstructed Skin, PLoS ONE 9(8): e105263, doi:10.1371/journal.pone.0105263, 2014.

**P2, L8: replace "depended" with "dependent"**
**P2, L11: replace "surface UV attenuation" with "attenuation of the solar UV radiation"**

Answer:

Suggested changes were made.

**P2, L10-12: add references to support your statement that "The absorption by SO2 (in the UV-B range) and NO2 (mostly in the UV-A range) is important for the surface UV attenuation only in extreme concentrations of such gases."**

Answer:

The sentence was changed to: "In the spectral range up to ~330 nm, absorption by ozone is usually much stronger than absorption by other main trace gases (SO2, NO2) (Cede et all, 2006)" (P2, L11-12). The reference is:
Cede, A., Herman, J., Richter, A., Krotkov, N., and Burrows, J.: Measurements of nitrogen dioxide total column amount using Brewer double spectrophotometer in direct Sun mode, J. Geophys. Res., 111, D05304, doi:10.1029/2005JD006584, 2006.

**P2, L12: Replace "surface intensity of UV" with "intensity of the solar UV radiation at the earth surface". Furthermore, in addition to the properties, the amount of aerosols and clouds also affect UV radiation.**
**P2, L12-14: Change the phrase: "The negative trends in these variables, found over many of the northern hemisphere mid-latitudinal sites in the 1989s and 1990s, lead to increases of both the UV-B and UV-A irradiance"**
**With**
**"Increases of both the UV-B and UV-A irradiance have been reported over several mid-latitudinal sites of the northern hemisphere since the beginning of the 1990s, which have been mainly attributed to decreasing attenuation by aerosols and clouds."**
**P2, L15: Replace "An" with "The".**
**P2, L15: Replace "the large urban agglomeration" with "large urban agglomerations".**
**P2, L17: Replace "UV cloudless sky irradiances" with cloudless-sky UV irradiances". Also replace "its suburbs" with "a sub-urban area near Athens".**
**P2, L19: Replace "The erythemal irradiance at the centre of Athens was 30% lower than at the suburbs with similar values of total ozone (TO3) for days with increased pollution in the air."**
**With**
**"The erythemal irradiance at the centre of Athens was up to 30% lower than at the outskirt site during days with increased air pollution over Athens basin and similar values of total ozone (TO3) over the two sites."**

Answer:

Suggested changes were made.

**P2, L20-21: What you write here is not clear. Please be more specific. Do you mean differences from the measurements or differences by corresponding simulations over the Athens basin?**

Answer:

The sentence was clarified (P2, L22-23): "A similar difference was noticed in the modelled UV-B irradiance with input from measurements of the total ozone (TO3) and aerosols optical depth (AOD) by the Brewer spectrophotometer (BS) at the outskirts of Athens.

**P2, L23: Delete "the" before "winter" and "summer". Replace "Mexico" with "Mexico City".**
**P2, L23-24: Are 9% and 21% the differences between the annual mean levels for winter and summer? Please be more precise.**
**P2, L24-25: " Corr et al. (2009) … 0.7 – 0.85". This sentence is not clear. Please rephrase.**
**P2, L26: Delete "atmospheric"**
**P2, L30: Delete "of aerosols"**

Answer:

Suggested changes were made.

**P3, L3-8: Notice that for a typical Angström parameter of ~1.5, the differences in AOD becomes ~2 times larger for UV-B wavelengths. I suggest that you should provide quantitative estimations of the changes in UV irradiance due to the reported differences in AOD (e.g. for a low SSA = 0.85) at this point, to prove that the reported differences in AOD do not induce large differences in UV irradiance. That can be easily achieved by performing modeling simulations. Furthermore, it should be mentioned that for organic particles, the absorption in the UV range may be even larger than that predicted by interpolating using the Angström parameter for the visible range of the spectrum (e.g. see Bais et al. (2015)\* and references therein).**
**\* Bais, A. F., R. L. McKenzie, G. Bernhard, P. J. Aucamp, M. Ilyas, S. Madronich, and K. Tourpali (2015), Ozone depletion and climate change: impacts on UV radiation,** *Photochemical & Photobiological Sciences*, *14*(1), 19-52, doi:10.1039/c4pp90032d.

Answer:

We performed numerical simulations of the ratio with the use of measured AOD for both sites in the revised manuscript (section 3.3). The reference was added on P3, L8-10:
"However, for organic particles, the absorption in the UV range may be larger than predicted using Angström parameters for the visible range of the spectrum (Bais et al., 2015).".

**P3, L4: Remove "," after "stated".**
**P3, L6: "the difference" instead of "it"**
**P3, L11: Delete "a specific"**
**P3, L14: Delete "at"**

**Methodology**
**P3,L23: Replace "at Belsk (51.8°N, 20.8°E, 190 m amsl), which is located in a rural region" with "Belsk (51.8°N, 20.8°E, 190 m amsl). The latter is a rural region"**
**P3, L25: Replace "the area" with "an area"**
**P3, L28: Replace "spectra accuracy" with "spectral accuracy"**

Answer:

Suggested changes were made.

**P4, L3: Please provide reference(s) regarding the fact that the estimated uncertainty in the erythemal irradiance is 5%.**

Answer:

The reference is:
Gröbner, J., and Schreder, J.: Protocol of the intercomparison at the Polish Geophysical Institute, Warsaw, Poland, May, 20‐22 2004 with the travelling standard spectroradiometer B5503 from ECUV within the project QASUME, http://www.pmodwrc.ch/wcc_uv/qasume_audit/reports/2004_05_poland_warsaw_PGI1.pdf, 2004.

**P4, L8: In this section you describe how each Brewer (207 and 64) is calibrated. Was the calibration procedure for the two Brewers the same before and after BS207 was moved to Warsaw? Is BS207 also calibrated against BS017? Please add some more information to convince the reader that the changes in the ratio are not due to a change in the calibration procedure or due to change of the BS207 characteristics during transportation from the one site to the other.**

Answer:

We added information about the BS207 calibrations on P4, L11-13:

"BS207 was calibrated against BS017 in 2012 and 2013. After the calibration in 2013, it was moved to Warsaw. Furthermore, it has been calibrated 3 to 4 times per year since 2010 with a set of standard lamps that allows elimination of instrument ageing (loss of its sensitivity to UVR)."

**P4, L11: Replace "The erythemal action spectrum follows CIE (1987)" with "The erythemal action spectrum is that suggested by the Commission internationale de l'éclairage (CIE) (CIE, 1987)"**

Answer:

The suggested change was made.

**P4, L13: Were there any criteria for the selection of the partly daily doses used for the comparison between Belsk and Warsaw? Is there a minimum amount of measurements (or measurements per 1 or 2 hours) below which the data are rejected? Calculation of the integral from only a small number of measurements and/or large gaps in the 3- or 6-hour period may lead to large differences between the integrals for the two sites. If not already done, I would suggest using a filter (e.g. use only time intervals with at least one measurement per 1 - 2 hours).**

Answer:

We used a filter to 10 measurements per day. BS064 takes measurements 3 times per hour, while BS207 takes measurements 2 times per hour. It changes only if one of the BSs does not work properly, and then the previous filter should be enough. We added the filter for specific time intervals and a few points were removed. Although, that did not have any impact on the ratios.

**P4, L15: How confident are you for your cloud detection method? Are there cloudy cases that cannot be detected? Can you estimate if, and in what extent, they affect your results?**

Answer:

The answer to this question was incorporated in the text on P4, L23-24:
"There is no strict mathematical criterion applied here, but rather an intuitive inspection of the time series shape.".

**P4, L23-30: The specific paragraph is carelessly written. Please try to re-write it more carefully.**
**Results**
**Figure 1: Add some more information in the manuscript for LOWESS filter and/or proper references.**
**P5, L8-9: I think that the phrase "The most of the differences lie within ±5% range" is not necessary here since in the previous paragraph the 1σ uncertainty (which by the way is 7%) is given.**

Answer:

Suggested changes were made. We added information and reference to LOWESS method on P4, L29-30: "The LOWESS (Locally Weighted Scatterplot Smoothing) filter (Cleveland, 1979) was used for smoothing of the curves.".
The reference is:
Cleveland W.S.: Robust Locally Weighted Regression and Smoothing Scatterplots, Journal of the American Statistical Association 74(368): 829-836, 1979.

**Figure 2a: In the specific figure there are two data points near 0.9. Is there any explanation for this large difference between the results from the two instruments for the particular days?**

Answer:

Yes, they were probably affected by clouds and were removed.

**P5, L22: Please specify that the higher latitude of the site at Warsaw means that for the same time, the solar zenith angle over Warsaw is always lower by ~0.5° compared to Belsk.**
**P5, L23: replace "surface albedo" with "different surface albedo"**
**P5, L25: replace "in" with "to perform"**

**P5, L27: Replace the phrase "prescribed values of surface albedo equal to 0.03 at Belsk and a set {0.03, 0.06, 0.12} in Warsaw" with "standard values of surface albedo equal to 0.03 for Belsk and 0.03, 0.06 and 0.12 for Warsaw"**
**P5, L28: Replace "of" before TO3 with "between" and delete "in"**
**P5, L30: Do you mean "coincidence" instead of "correspondence"? In this case it is for the entire range of the TO3 variability of both sites and not only of Belsk.**
**P6, L1: "assuming that" instead of "assuming"**

Answer:

Suggested changes were made.

**Figure 4: There is an obvious annual cycle of the ratio. I suppose that this is due to the stronger effect of the 0.5° difference in SZAs in winter, when the SZAs are larger. Thus, the effect of larger albedo compensates for the effect of different latitude only for a specific period of the year. The same annual cycle is obvious in Figure 5, possibly due to the remaining effect of the difference in SZA. Given that in Figure 4 there are no results for December and January, when the effect of SZA is expected to be even larger, while in Figure 5 there are results for the particular months, I believe that the deviation of the mean ratio from unity is partially due to the effect of different SZAs. I suggest that you should either quantify the remaining effect due to different SZAs and take it into account in the discussion of the results presented in Figures 5 and 6, or alternatively compare the irradiances for specific SZAs (thus slightly different time).**

Answer:

Yes, we do agree with the Referee #1. We restructured the modelling section (now section 3.3) and performed a set of numerical simulations with LibRadtran. The dependence of the ratio on SZA was shown in Figure 6.

**P6, L11: remove "of"**
**P6, L11: remove "in"**
**P6, L15: replace**
**"in the periods symmetrical around local noon for 6h for all-sky and 3h before noon for cloudless sky conditions"**
**with**
**"for 6h symmetrical periods around local noon for all-skies, and 3h periods before local noon for cloudless skies"**
**P6, L17: Delete "previously"**
**P6,L19: Replace "The" with "A"**

Answer:

Suggested changes were made.

**P6,L23: It seems to me that the ratio oscillates around ~1.05 and not 1. As already commented, I believe that part of the spread in the calculated ratios is due to the remaining effect of the difference in the latitude of the two sites.**

Answer:

Yes, we do agree with the Referee #1. It was changed on P6, L15: "The ratio oscillates around 1.05 within the range between 0.9 and 1.2.".

**Figure 5: I suppose that the slightly different pattern of the temporal evolution of the ratios for the erythemal doses and the UV-A1 doses are again because of the effect of the different SZA. The effect of different SZAs is stronger for lower wavelengths, being partially responsible for the larger ratios of the erythemal doses compared to those of the UV-A1 doses.**

We do agree with the Referee #1.

**P7, L6: "attenuates" instead of "attenuate"**

**P7, L10: "emissions" instead of "emission"**

Answer:

Suggested changes were made.

**P7, L11: add references to support your statement that "causing numerous cases over the EU air quality threshold"**

Answer:

The reference was added on P7, L27-28: "(...) causing numerous cases over the EU air quality threshold (Monitoring System of Air Quality in Mazowieckie Region, http://sojp.wios.warszawa.pl/).".

**P7, L12: what do you mean with the phrase "makes specific boundary layer"? Please explain (e.g. is the upper limit of the boundary layer higher compared to the nearby rural areas?).**
**P7, L12-13: Add references.**

Answer:

The explanation and references were added on P7, L29-31:
"(...) i.e. in the boundary layer factors like wind, temperature, moisture, turbulence and energy budget fields differ from nearby rural sites (e.g. Fortuniak et al., 2005, Miao et al., 2009, Haberlie et al., 2015).".

**P7, L15: "higher AOD at 500 nm over Warsaw" instead of "higher Warsaw AOD values at 500 nm"**
**P7, L16: "similar differences" instead of "similar values"**
**P7, L21:"~2% more attenuation" is more accurate than " ~2% attenuation".**

Answer:

Suggested changes were added.

**P7, L21-24: Again, these numbers may change after re-evaluation of the effect of the SZA.**

Answer:

Re-evaluated numbers were added on P8, L2-7:
"(...) the Belsk/Warsaw ratio between the erythemal and UV-A (324 nm) doses is ~1.06 and ~1.04, whereas the ratio is ~1.08 and ~1.06 for all-sky conditions, respectively. The aerosol effects are responsible for ~2% larger erythemal and UV-A near-noon doses at Belsk. The cloud effects add 2%, enlarging the Belsk-Warsaw difference. The SZA effects due to the longitudinal/latitudinal difference between the sites lead to 3% (or 2%) greater erythemal (or UV-A) doses at Belsk. The difference is even larger in the cold period of the year (for higher SZAs). The unexplained 1% higher doses at the rural site for the erythemal doses ratio could be attributable to instrument issues.".

**P7, L30-31: "An Indirect method for BS has been proposed by Bais et al. (2005)" instead of "Indirect method for BS has been proposed (Bais et al., 2005)"**

Answer:

The suggested change was made.

**P8, L2: add appropriate reference to support that the typical SSA for rural sites is 0.92.**

Answer:

The explanation was added on P5, L7: "(...) SSA=0.92, which is a mean value measured by the CIMEL sunphotometer at Belsk (...)".

**P8, L2-4: Since the overall difference is 6% and AOD difference is responsible for 2% (according to what is written in the previous paragraph) then SSA differences should compensate for 4% - and not 5% - of the difference. However, this might be different if you take into account the effect of the SZA.**

Answer:

The discussion about SSA and AOD effect was changed, after taking into account the effect of SZA. It is in section 4 (P8, L8-18):
"It seems possible that urban aerosols lead to higher absorption of the UV irradiance, i.e. small SSA values (<0.9) could characterise such aerosols. On the other hand, the albedo of urban surfaces is higher in the snowless period, that may compensate the effects of lower urban aerosols' SSA. Analysing the UV radiation in the Mexico City metropolitan area, Castro et al. (2001) found the urban albedo of 0.12 over asphalt and grey surface cement sites. This is four times larger than the commonly used albedo of 0.03 over grass. Parisi et al. (2004) found that over some non-shaded parts of the city with high albedo (e.g. concrete surface) there is an amplification of the human exposure of up to 7% for people in the upright position. We performed RTM simulations with the observed TO3 and AOD values over Warsaw to fully compensate (by absorbing aerosols) the UV increase due to changes in albedo from 0.03 to 0.12. SSA=0.86 and 0.85, for SZA=60° and 30°, respectively, are found for the city site, i.e., 0.06 and 0.07 less than the value previously used in our RTM simulations for rural aerosols. Such estimate looks probable, as the Warsaw observing site is among the most polluted parts of the city because of abnormal vehicle emissions in the nearby main city roads.".

**P8, L5-7: I suggest moving the sentence "Kazadzis et al. (2009b) … quality there" to P7, L27, after "… to SSA changes."**
**P8, L6: "in Thessaloniki" instead of "in the Thessaloniki"**

Answer:

Suggested changes were made.

**P8, L20 – 31: As already commented, the effect of difference in SZA is more pronounced during the cold period (larger SZAs) and less pronounced during the warm period (smaller SZAs). The effect of different SZAs has to be removed – or taken into account - properly, so that you can get more accurate conclusions.**

Answer:

It was reconsidered.

**Figure 8: I recommend adding a paragraph (e.g. 3.4) and expand the discussion relative to Figure 8. Furthermore, the discussion regarding the agreement of your results with the results of other recent studies (e.g. Zerefos et al. (2012), de Bock et al. (2014), Fountoulakis et al. (2016)), as well as the reasons for this agreement could be expanded.**

Answer:

We have decided to remove this figure and the part of the discussion as it is not connected with the Belsk-Warsaw comparison.

**P8, L24 - 26: "Thus, it seems possible that increased cloudiness over urban areas does not necessarily mean increased attenuation of solar radiation, since modification of the cloud structure and properties by the urban aerosols may lead to the formation of clouds which attenuate the solar radiation less effectively" instead of**
**"Thus it seems possible that even higher cloudiness over urban areas does not mean higher attenuation of solar radiation, because the urban aerosols modify the cloud structure compensating the effect of increased cloud cover there"**
**P8, L27: "the" instead of "an"**
**P8, L32: "level" instead of "levels" and "higher than in the past" instead of "high"**

Answer:

Suggested changes were made.

**P8, L33: Zerefos et al. (2012) discuss the trends of the UVR after the mid-1990s. Thus, in the particular study they do not discuss the increase of the UVR due to the decrease of ozone until the mid-1990s.**
**P9, L2-3: Add proper reference to support your statement that that the environmental pollution was enormous in the mid-1970 and early-1980. Furthermore, the UV increases because aerosol and clouds decrease relative to the past – not because they were high in the past.**
**P9, L6: I do not think that 5-8% is "slightly" lower. I suggest removing the phrase "only slightly".**

Answer:

This part of the manuscript was removed. The long-term trend is out of main scope of the revised manuscript.

**P9, L8: "parts" instead of "part"**

Answer:

Suggested change was made.

---

## Author Comment (AC3) · 7 Oct 2016

**General remarks**

**The authors compare broad-band solar UV radiation exposure data derived from measurements by Brewer spectrometers taken in the city of Warsaw and outside the city. The paper is logically separated into sections. The abstract gives an overview of the paper. It mentions its most significant results. The SI is used throughout the paper. The title uses 'UV exposure', though 'UV dose' is used in the text. The same name should be used in the title and in the text. The number of Figures is adequate to illustrate the text. However, as will be addressed in more detail below, Figure captions need to be clear and provide sufficient details on what is presented. If a Figure consists of two graphs, the Figure caption should address both parts separately, for example by numbering them as a) and b). The axis names should be completed to mention the parameters in the graph.**

Answer:

The title and figures were changed.

**It is recommended to apply corrections to English grammar and language style of the manuscript, and also correct for the many writing errors.**

Answer:

The corrections were made. The manuscript was significantly improved.

**The method of separation between the albedo effect on the one hand, and the aerosol and cloud effects to UV exposure at the sites on the other hand is not clear. The authors assume that the albedo is 3% at Belsk and 6% at Warsaw. This difference of surface albedo is shown by model calculations to compensate for the difference in solar height (or latitude) between the sites, if the aerosol load would be the same. The remaining differences in the UV measurements are then interpreted as a result of differences in aerosol loads and cloudiness between the sites. Have you checked the real albedo at the sites? How about seasonal differences in albedo for example due to snow cover?**

Answer:

We modeled values for different albedos in Warsaw only for the period without snow cover, because in that period UV radiation is the strongest and can cause sunburns or DNA damage. The values of albedo used for modeling refers to the article of Castro et al. (2001) – for Belsk albedo is 0.03 (green grass) and for Warsaw we examined a set of albedo values 0.03, 0.06 and 0.12 (reference have been added to the references list). Such albedo values are taken into consideration, because in the measuring site the terrain is a mixture of surfaces – from green grass to asphalt. We did not check the real albedo at the sites. The effect of seasonal differences in albedo was discussed in the previous version of the manuscript (last paragraph of section 3.2 and sixth paragraph of section 4). It was removed from current version of the manuscript, as the main seasonal difference is found to be the effect of the difference in latitude between the sites.

**Detailed comments**

**Page 1, line 20: 'increase of UV exposure for peoples' replace by 'higher UV exposure for people'**
**Page 3, Section 2: Coordinates of the sites and the types of the immediate surroundings should be included here, not only mentioned in the abstract.**
**Page 3, line 8: 'Its'**
**Page 3, line 10: replace 'diffusive' by 'diffuse'**
**Page 4, line 17: replace 'moment' by 'time period'**
**Page 6, line 11: replace 'an increase of BS064/BS207' by 'a higher BS064/BS207'**
**Page 6, line 31: replace 'decline' by 'difference'**
**Page 7, line 7: 'radiation'**

Answer:

Suggested changes were made.

**Page 3, second paragraph: Is the higher contribution of diffuse irradiance to global irradiance the cause of the higher internal stray light of instruments, or is it just the lower global irradiance?**

Answer:

The higher contribution of diffuse irradiance to global irradiance is the cause of the higher internal stray light of instruments.

**Page 3, fourth paragraph: Do you really mean 'clear sky conditions' that refer to no aerosol or low aerosol load, or do you mean 'cloudless conditions'? If you refer to the latter, the changed wording needs to be applied to the whole text.**

Answer:

We meant 'cloudless conditions'. Changes in the text were made.

**Conditions of cloudless sky (or clear sky) were separated from the relative increase of irradiance over time around noontime. Using only this criterion, the separated 'cloudless cases' may still contain cases of clouds that do not occlude the sun. Have you checked by cloud observations or cloud imaging data, how good your selection criterion to find real cloudless cases is?**

Answer:

We checked all separate spectras by observing the shape of the UV irradiance curves. The day was omitted, when the shape was different from the "bell" shape and values of the UV irradiance were far from expected.

**Page 3, line 28: You refer to the erythemal action spectrum by CIE (1987). Probably, you have taken into account the corrections, as discussed by Webb et al. (2011), Photochem. Photobiol. 97, 483 – 486. If so, the citation should be added.**

Answer:

We used the original action spectrum by CIE (1987) without corrections. Uncertainties between values weighted with different action spectra are not more than 2%. Taken into consideration that we calculated ratios, uncertainties are even smaller and comparable with the measurement error.

**Page 5, last paragraph: You state that the main cause of scatter in the interpolated UV irradiance values is the first and last spectrum of the time period. Why did you not leave those two spectra?**

Answer:

This issue was clarified in the manuscript: "The main reason for this scatter is the interpolated erythemal (or UV-A1) irradiance value at the beginning (3.5 h before local noon) and at the end (local noon-0.5h) of the calculated period. BS observations were rarely made exactly at the starting and ending moments. Thus linear interpolated values were used taken from observations closest to the beginning or to the end of the period, i.e. the irradiance values just outside the observing period were taken into account."

**Figure captions are incomplete and partly confusing. The caption of Fig. 3 says 'The same as Fig. 1', but the ratios are calculated for total ozone values measured simultaneously at Belsk and Warsaw'. Does the upper part of Fig. 3 show ratios of erythemal exposure? Is it measured or modelled? Or, does it show ratios between measured column ozone at the sites? The vertical axis only states 'Belsk/Warsaw'. The lower panel does obviously show total ozone at the sites, but it is not mentioned in the Figure caption. Caption of Fig. 4 should mention that it refers to 'modelled ratios'. Figure 5 should mention that it refers to 'measured ratios'.**

Answer:

The captions and vertical axis names were changed.

**An urban agglomeration effect on surface UV doses: Comparison of the Brewer measurements in Warsaw and at Belsk, Poland, for the period 2013-2015**

Agnieszka E. Czerwińska[1], Janusz W. Krzyścin[1], Janusz Jarosławski[1]

[1] Institute of Geophysics, Polish Academy of Sciences, Warsaw, 01452, Poland

*Correspondence to*: Agnieszka E. Czerwińska (aczerwinska@igf.edu.pl)

**Abstract.** The specific aerosols and cloud properties over large urban regions seem to generate an island, similar to the well-know city heat island, with lower intensity of the UV radiation compared to the surrounding cleaner areas, thus creating a shield against excessive human exposure to the UV radiation. The present study focuses on differences in the erythemal and UV-A1 (340-400 nm) doses measured by the Brewer spectrophotometers in Warsaw (52.3ºN, 21.0ºE) and at Belsk (51.8ºN, 20.8ºE), which is located in a rural region at a distance of about 60 km in the south-west direction from the city. The ratio between  erythemal and UV-A1 partly daily doses, obtained during all-sky and cloudless-sky conditions in the period May 2013-December 2015, are analyzed to infer specific clouds and aerosols forcing on the surface UV doses over Warsaw. Radiative model simulations are carried out to assess impact of the Warsaw-Belsk  differences in total ozone, geographical location, and albedo on the mean ratio between the doses. Higher surface albedo over the city compensates the effect of its more northern location as the mean total ozone values appear almost the same over both sites. It is found that urban agglomeration induced 8% and 5% attenuation of  the erythemal and UV-A1 doses, respectively, which could be caused by larger UV absorptions by the urban aerosols  It appears that a slightly increased optical depth of the urban aerosols and properties of clouds generated over Warsaw are less important for the UV attenuation. Taking into account previously found ~10%` higher UV exposure for people in the upright position due to higher albedo of the city surfaces, it could be hypothesized that the urban pollution in Warsaw does not create a shield against incoming  UV radiation.

[revised manuscript text omitted]

---

## Author Comment (AC5) · 7 Oct 2016

[revised manuscript text omitted]
 atmosphereand clouds over Warsaw cannot be treated as ado not provide an effective shield against excessive human exposureUVR.

**Acknowledgements**. This work was partially supported withinby statutory activities No. 3841/E-41/S/2016 of the Ministry of Science and Higher Education of Poland. We appreciate a support from early-carrier scientist grant (A. Czerwińska) No. 500-10-18 by the Institute of Geophysics, Polish Academy of Sciences.

**5    References**

Acosta, L.R., and Evans, F.J.: The design of the Mexico City UV monitoring network: UV-B measurements at ground level in the urban environment, J. Geophys. Res., 105, 5017–5026, 2000

Bais, A.F., Kazantzidis, A., Kazadzis, S., Balis, D.S., Zerefos, C.S., and Meleti, C.: Deriving an effective aerosol single scattering albedo from spectral surface UV irradiance, Atmos. Env., 39, 1093-1102, 2005.

Bais, A. F., R. L. McKenzie, G. Bernhard, P. J. Aucamp, M. Ilyas, S. Madronich, and K. Tourpali, Ozone depletion and climate change: impacts on UV radiation, Photochemical & Photobiological Sciences, 14(1), 19-52, doi:10.1039/c4pp90032d, 2015.

Castro, T., Mar, B., Longoria, R., Ruiz-Suárez, L. G., &and Morales, L. (2001).: Surface albedo measurements in Mexico City   metropolitan area. Atmósfera, 14(2), 69-74. Recuperado en 10 de junio de 2016, de          .
http://www.scielo.org.mx/scielo.php?script=sci_arttext&pid=S0187-  62362001000200002&lng=es&tlng=en, 2001.

Cede, A., Herman, J., Richter, A., Krotkov, N., and Burrows, J.: Measurements of nitrogen dioxide total column amount using Brewer double spectrophotometer in direct Sun mode, J. Geophys. Res., 111, D05304, doi:10.1029/2005JD006584, 2006.

Corr, C. A., Krotkov, N., Madronich, S., Slusser, J. R., Holben, B., Gao, W., Flynn, J., Lefer, B., and Kreidenweis, S. M.: Retrieval of aerosol single scattering albedo at ultraviolet wavelengths at the T1 site during MILAGRO, Atmos. Chem. Phys., 9, 5813-5827, doi:10.5194/acp-9-5813-2009, 2009.

Chubarova, N.Y., Sviridenkov, M.A., Smirnov, A., and Holben B.N.: Assessments of urban pollution in Moscow and its radiative effects, Atmos.Meas.Tech., 4, 367-378, doi:10.5194/amt-4-367-2011, 2011.

CIE: A reference action s,pectrumspectrum for ultraviolet induced erythema in human skin, ed. A. F. MacKinley iand B. L. Diffey, _CIE ——J., 6((1)), 17–22, 1987.

Cleveland W.S.: Robust Locally Weighted Regression and Smoothing Scatterplots, Journal of the American Statistical Association 74(368): 829-836, 1979.

Daniel, W. W., and Cross, C.L.: Biostatistics: a foundation for analysis in the health sciences. New York: John Wiley and Sons, 2013.

De Bock, V., De Backer, H., Van Malderen, R., Mangold, A., and Delcloo, A.: Relationship between erythemal UV dose, global solar radiation, total ozone column, and aerosol optical depth at Uccle, Belgium, Atmos. Chem. Phys., 14, 12251-12270. doi: 10.5194/acp-14-12251-2014, 2014.

Engelsen, O., and Kylling, A.: Fast simulation tool for ultraviolet radiation at the Earth's surface, Opt. Eng., 44(4), 041012, doi:10.1117/12.639087, 2005.

Fioletov, V.E., J. B. Kerr, C. T.J.B. , McElroy, D. I.C.T., Wardle, V.D.I., Savastiouk, V., and T. S. 
[revised manuscript text omitted]

[Figure]

[Figure]

[Figure]

Figure 68: Yearly means of the : The Belsk/Warsaw ratio between erythemal dose measured (a) and UV-A (324nm) (b) irradiances calculated by the libRadtran model for 2015 versus SZA at Belsk for the period 1976-2015 at Belsk (squares10:40 (GMT). The solid curve denoterepresents the smoothed data by LOWESS filter.

[Figure]

**Figure 7a:** **The ratio between total ozone values measured by the Brewer Spectrophotometer No. 64 and No. 207 while working simultaneously at Belsk and in Warsaw for the period May 2013-December 2015. The solid curve represents the smoothed data by LOWESS filter.**

**Figure 7b: Scatter plot of total ozone values measured by the Brewer Spectrophotometers No. 207 and No. 064 while working simultaneously at Belsk and in Warsaw from May 2013 to December 2015.**

[Figure]

**Figure 8a: The ratio between AOD at 550nm measured simultaneously by MODIS over Belsk and Warsaw in the period May 2013-December 2015. The solid curve represents the smoothed data by LOWESS filter.**

**Figure 8b: Scatter plot of AOD at 550nm measured by MODIS over Belsk and Warsaw from May 2013 to December 2015.**

---

## Editor Comment (EC2) · S. Kazadzis (Editor) · 11 Oct 2016

I would like to include here some comments on the revised manuscript.

Page 2, line 14, clouding – clouds

P 2, L31: trends – positive trends

P3 L16: "We will strive to support (or disprove) the hypothesis by comparing the erythemal and UV-A (324 nm) radiation measurements by the BSs in Warsaw and Belsk for the period May 2013-December 2015."

I think this can be eliminated as at this point as you can clearly say if this hypothesis is correct or not.

Page 4 , L2 stray light needs a reference.

P4, L28 the same ratio – Which one (wavelengths/erythemal) ?

P5 line 8-13

Since there are other publication that are showing significant differences of UV ssa compared with the visible one especially at urban areas I would suggest to change the paragraph (and remove non used references after that) :

"We used SSA at 440 nm as a constant for the whole ultraviolet spectrum, as it was found that monthly averages estimated from BS at Uccle were in close agreement with the CIMEL measurements at 440 nm, especially for 320 nm (Nikitidou et al., 2013). Furthermore, Liu et al. (1991) performed Mie calculations for the rural aerosol model (Shettle and Fenn, 1979) and suggested that for this type of aerosol, SSA is approximately independent of wavelength. There are no measurements performed for SSA at the UV wavelength range."

To:

Since there are no AERONET related measurements of SSA at UV wavelengths , we used SSA at 440 nm as a constant for the whole ultraviolet spectrum, as it was found that monthly averages estimated from BS at Uccle were in close agreement with the CIMEL measurements at 440 nm, especially for 320 nm (Nikitidou et al., 2013).

Also, because the suggestion here that SSA is independent of wavelength is in contradiction with your discussion hypothesis of SSA can be lower in the UV.

P5 line 20 The mean ratio of which wavelength range (eryhthemal)?

P6 line 8 : (local noon - 3h, local noon-0.5h) is not 3 hours

Figure 8a: AOD ratios are misleading in this case. In addition, absolute AOD differences are related with changes in solar radiation and not their ratio. I would suggest to put AOD differences instead and change the text accordingly.

I would suggest to include a table in the end of section 3 including all mean cloudless sky ratios and standard deviations for all factors analyzed (intercomparison, solar angle, ozone, AOD, actual ratios). In order to summarize the quantification of all effects.

I still think that the latitude difference of the two stations (solar zenith angle effect) as also pointed out from the reviewers can be eliminated. This is because including it to the factors affecting the differences among the sites introduces an uncertainty as it is changes from day to day and in the end in terms of percentage is the most important difference.

This can be done by either normalizing the irradiance of one of the stations using the solar zenith angel functions and compare them again. Or, as suggested, use ratios of measurements (and not 3 or 6 hour averages) for certain solar zenith angle windows e.g. X±1 degrees where X can be e.g. 45 – 60 – 75 degrees. (75 degrees will capture the whole year). Then even if the measurements correspond to different time for the two stations, they are only slightly affected by the solar zenith angle issue.

conclusions

As you write (e.g. for the erythemal) you have (roughly) a 6% difference that can be attributed 3-4% on the different solar angles, 1% on the instrument differences and 2% to aerosol difference. So more or less everything is explained. Thus in the paragraph describing albedo and SSA you are mentioning two hypothetical (there are no measurements) suggestions (a: albedo might be higher in Warsaw site and b. SSA might be lower). I would suggest rewriting this paragraph mostly suggesting that these two parameters (albedo and SSA); a. has been just assumed, b. they can be different and c. there is a possibility that (based on the modeling calculations) the effect of the one is masking the effect of the other. All the above, having in mind that this is a

discussion that is not based in actual measurements.

"Our study proves that the UV level in Warsaw is slightly lower than that found in cleaner suburbs of the city. Thus urban aerosols and clouds over Warsaw do not provide an effective shield against excessive UVR"

I would change that to "Our study proves that the UV level in Warsaw is slightly lower than that found in cleaner suburbs of the city. The differences that were attributed due to AOD differences are in the order of the accuracy of the instruments used. Based on the Brewer measurements, urban aerosols and clouds over Warsaw only partially act as an effective shield against excessive UVR.

In addition, it would be interesting to try to justify this conclusion.
* * *

---

## Author Comment (AC6) · 13 Oct 2016

**Response to Editor's (Stelios Kazadzis) comments**

**I would like to include here some comments on the revised manuscript.**

**Page 2, line 14, clouding – clouds**
**P 2, L31: trends – positive trends**
**P3 L16: "We will strive to support (or disprove) the hypothesis by comparing the erythemal and UV-A (324 nm) radiation measurements by the BSs in Warsaw and Belsk for the period May 2013-December 2015."**
**I think this can be eliminated as at this point as you can clearly say if this hypothesis is correct or not.**

Answer:

The suggested changes were made.

**Page 4 , L2 stray light needs a reference.**

Answer:

The following reference was added to the manuscript:
Bais, A. F., Zerefos, C. S., McElroy, C. T.: Solar UVB measurements with the double- and single-monochromator Brewer ozone spectrophotometers, Geophys. Res. Lett., 238, 833–836, doi: 10.1029/96GL00842, 1996.

**P4, L28 the same ratio – Which one (wavelengths/erythemal) ?**

Answer:

This issue was clarified in the paper, P4, L24-29: "Ratios between erythemal and UV-A (324 nm) doses (...) The same ratios are measured for the period of the Warsaw observations (May 2013 to December 2015) by BS207 and BS064 at Belsk to assess the impact of the urban agglomeration on the erythemal and UV-A radiation."

**P5 line 8-13**
**Since there are other publication that are showing significant differences of UV ssa compared with the visible one especially at urban areas I would suggest to change the paragraph (and remove non used references after that) :**
**"We used SSA at 440 nm as a constant for the whole ultraviolet spectrum, as it was found that monthly averages estimated from BS at Uccle were in close agreement with the CIMEL measurements at 440 nm, especially for 320 nm (Nikitidou et al., 2013). Furthermore, Liu et al. (1991) performed Mie calculations for the rural aerosol model (Shettle and Fenn, 1979) and suggested that for this type of aerosol, SSA is approximately independent of wavelength. There are no measurements performed for SSA at the UV wavelength range."**
**To:**
**Since there are no AERONET related measurements of SSA at UV wavelengths , we used SSA at 440 nm as a constant for the whole ultraviolet spectrum, as it was found that monthly averages estimated from BS at Uccle were in close agreement with the CIMEL measurements at 440 nm, especially for 320 nm (Nikitidou et al., 2013).**
**Also, because the suggestion here that SSA is independent of wavelength is in contradiction with your discussion hypothesis of SSA can be lower in the UV.**

Answer:

The suggested change was made.

**P5 line 20 The mean ratio of which wavelength range (eryhthemal)?**

Answer:

We meant erythemal doses. It was changed to "The mean value of the ratio between erythemal doses (...)" (P5, L19-20).

**P6 line 8 : (local noon - 3h, local noon-0.5h) is not 3 hours.**
Answer:

It should be "(local noon -3.5h, local noon -0.5h)" and was corrected.

**Figure 8a: AOD ratios are misleading in this case. In addition, absolute AOD differences are related with changes in solar radiation and not their ratio. I would suggest to put AOD differences instead and change the text accordingly.**

Answer:

The suggested change was made. Figure 8a and according text was corrected.

**I would suggest to include a table in the end of section 3 including all mean cloudless sky ratios and standard deviations for all factors analyzed (intercomparison, solar angle, ozone, AOD, actual ratios). In order to summarize the quantification of all effects.**

Answer:

All cloudless sky ratios and their standard deviations were included in Table 1.

**I still think that the latitude difference of the two stations (solar zenith angle effect) as also pointed out from the reviewers can be eliminated. This is because including it to the factors affecting the differences among the sites introduces an uncertainty as it is changes from day to day and in the end in terms of percentage is the most important difference.**
**This can be done by either normalizing the irradiance of one of the stations using the solar zenith angel functions and compare them again. Or, as suggested, use ratios of measurements (and not 3 or 6 hour averages) for certain solar zenith angle windows e.g. X− 1 degrees where X can be e.g. 45 – 60 – 75 degrees. (75 degrees will capture**
**the whole year). Then even if the measurements correspond to different time for the two stations, they are only slightly affected by the solar zenith angle issue.**

Answer:

We calculated ratios according to Editor's comment. The results are in Table 1 and on P7, L8-12:
"To eliminate the SZA's effect on the ratios, we calculated also mean irradiances ratios for specified SZA windows for cloudless conditions. Calculations were done for SZA windows: $45°\pm1°$, $60°\pm1°$ and $75°\pm1°$. For erythemal irradiances, the ratios were $1.02 \pm 0.05$ ($1\sigma$), $1.03 \pm 0.04$ ($1\sigma$) and $1.02 \pm 0.05$ ($1\sigma$), respectively. For UV-A (324 nm) irradiances, the ratios were $1.02 \pm 0.05$ ($1\sigma$), $1.02 \pm 0.04$ ($1\sigma$) and $1.01 \pm 0.04$ ($1\sigma$)."
We also added this result into discussion on P8, L5-8:
"The aerosol effects are responsible for ~2% larger erythemal and UV-A near-noon doses at Belsk, which stays in agreement with calculations of irradiances ratios between the sites for specified SZA windows ($45°\pm1°$, $60°\pm1°$ and $75°\pm1°$). After eliminating the SZA's effect for cloudless-sky conditions, both erythemal and UV-A (324 nm) irradiances at Belsk were ~2% higher than in Warsaw."

**conclusions**
**As you write (e.g. for the erythemal) you have (roughly) a 6% difference that can be attributed 3-4% on the different solar angles, 1% on the instrument differences and 2% to aerosol difference. So more or less everything is explained. Thus in the paragraph describing albedo and SSA you are mentioning two hypothetical (there are**
**no measurements) suggestions (a: albedo might be higher in Warsaw site and b. SSA might be lower). I would suggest rewriting this paragraph mostly suggesting that these two parameters (albedo and SSA); a. has been just assumed, b. they can be different and c. there is a possibility that (based on the modeling calculations) the effect of the one is masking the effect of the other. All the above, having in mind that this is a discussion that is not based in actual measurements.**

Answer:

The paragraph was re-written to: "(...) We performed RTM simulations to show that the effect of higher surface albedo in Warsaw (the UV irradiances increase) can be compensated by lower values of SSA. We did not measure surface albedo and SSA values. Thus, we assume that the surface albedo in Warsaw can be in the range of 0.03 to 0.12 and 0.03 at Belsk. We also assume, that SSA at Belsk is 0.92, which is a mean value measured by CIMEL photometer at 440 nm. For calculations, we used observed $TO_3$ and AOD values over Warsaw. SSA=0.86 and 0.85, for SZA=60º and 30º, respectively, were found for the city site, i.e., 0.06 and 0.07 less than the value previously used in our RTM simulations for rural aerosols. Such estimate looks probable, as the Warsaw observing site is located in the most polluted part of the city because of high vehicle emissions from the nearby main city road."

**"Our study proves that the UV level inWarsaw is slightly lower than that found in cleaner suburbs of the city. Thus urban aerosols and clouds over Warsaw do not provide an effective shield against excessive UVR"**
**I would change that to "Our study proves that the UV level in Warsaw is slightly lower than that found in cleaner suburbs of the city. The differences that were attributed due to AOD differences are in the order of the accuracy of the instruments used. Based on the Brewer measurements, urban aerosols and clouds over Warsaw only partially act as an effective shield against excessive UVR.**
**In addition, it would be interesting to try to justify this conclusion.**

Answer:

The last paragraph was rephrased following the Editor's suggestion. In addition, we added a justification of our conclusion on P9, L24-27: "For example, for UV index 5, time needed to get 1 MED (minimum erythema dose) for the person with phototype II is 33 minutes and for phototype III is 40 minutes (Fitzpatrick, 1988). Taking into consideration the attenuation of erythemal irradiances by 4%, which is the summarised effect of aerosols and clouds in Warsaw, this time for both phototypes changes only by 2 minutes. This small difference is not significant for planning and executing routine daily activities."

---

## Editor Comment (EC3) · S. Kazadzis (Editor) · 14 Oct 2016

I still have two comments that could potentialy improve the manuscript based on this last versions

Frist issue:

"Such estimate looks probable, as the Warsaw observing site is located in the most polluted part of the city because of high vehicle emissions from the nearby main city road."

This suggestion is very tricky as Warsaw and Belsk have similar AOD. So heavy pollution in Warsaw would have an effect on AOD too and not only on SSA.

So I would suggest: Thus, we assume that the surface albedo in Warsaw can be in the range of 0.03 up to 0.12 and 0.03 at Belsk. If we also assume, that SSA at Belsk is 0.92, which is a mean value measured by CIMEL photometer at 440 nm, using RTM we calculate an SSA down to 0.86 and 0.85, (for SZA=60° and 30°, respectively), for the maximum albedo difference. Such hypothesis could be only backed up with additional aerosol absorption measurements at the two sites.

second issue: In my comment:

"Our study proves that the UV level in Warsaw is slightly lower than that found in cleaner suburbs of the city. The differences that were attributed due to AOD differences are in the order of the accuracy of the instruments used. Based on the Brewer measurements, urban aerosols and clouds over Warsaw only partially act as an effective shield against excessive UVR. In addition, it would be interesting to try to justify this conclusion.

By justifying this conclusion I did not mean presenting the impact of such an effect (UV Index analysis). I mean are there any suggestions for having similar AOD at Belsk and Warsaw ? for example any additional information or assumptions about possible ouflow of aerosols from Warsaw ro Belsk ? any other aerosol source that could affect both locations ? any aerosol removal mechanisms present in Warsaw ? or something else ?

---

## Author Comment (AC9) · 19 Oct 2016

**I still have two comments that could potentialy improve the manuscript based on this last versions**

**Frist issue:**
**"Such estimate looks probable, as the Warsaw observing site is located in the most polluted part of the city because of high vehicle emissions from the nearby main city road." This suggestion is very tricky as Warsaw and Belsk have similar AOD. So heavy pollution in Warsaw would have an effect on AOD too and not only on SSA.**
**So I would suggest: Thus, we assume that the surface albedo in Warsaw can be in the range of 0.03 up to 0.12 and 0.03 at Belsk. If we also assume, that SSA at Belsk is 0.92, which is a mean value measured by CIMEL photometer at 440 nm, using RTM we calculate an SSA down to 0.86 and 0.85, (for SZA=60 and 30, respectively), for the maximum albedo difference. Such hypothesis could be only backed up with additional aerosol absorption measurements at the two sites.**

Answer:

The proposed change will be made.

**second issue: In my comment:**

**"Our study proves that the UV level in Warsaw is slightly lower than that found in cleaner suburbs of the city. The differences that were attributed due to AOD differences are in the order of the accuracy of the instruments used. Based on the Brewer measurements, urban aerosols and clouds over Warsaw only partially act as an effective shield against excessive UVR. In addition, it would be interesting to try to justify this conclusion.**
**By justifying this conclusion I did not mean presenting the impact of such an effect (UV Index analysis). I mean are there any suggestions for having similar AOD at Belsk and Warsaw ? for example any additional information or assumptions about possible ouflow of aerosols from Warsaw ro Belsk ? any other aerosol source that could affect both locations ? any aerosol removal mechanisms present in Warsaw ? or something else ?**

Answer:

We would like to remove UV Index analysis and change the last paragraph to:
„Our study proves that the UV level in Warsaw is slightly lower than that found in cleaner suburbs of the city. The differences that were attributed due to AOD differences are in the order of the accuracy of the instruments used. In Warsaw the dominant wind direction was from west, south-west (all-sky conditions) and east, south-east (clear-sky) in the period 2013-2015, so Warsaw did not have an impact on AOD at Belsk. Furthermore, Pietruczuk (2013) found that the advection of air masses to Belsk is mostly from the westerly direction. However, despite the fact that Warsaw is one of the most air-polluted city in Poland, AOD in Warsaw could be lowered by existing city ventilations paths (City of Warsaw, 2006). Most of this paths seem to generate micro-advection from nearby forests and parks. Based on the Brewer measurements, urban aerosols and clouds over Warsaw only partially act as an effective shield against excessive UVR."

Additional references:
City of Warsaw: Ecophysiographic Study for Purposes of the Study of Conditions and Directions of Spatial Management (http://architektura.um.warszawa.pl/ekofizjografia#do_pobrania), 2006 [In Polish].
Pietruczuk, A.: Short term variability of aerosol optical thickness at Belsk for the period 2002–

2010, Atmospheric Environment, Volume 79, Pages 744-750, ISSN 1352-2310, http://dx.doi.org/10.1016/j.atmosenv.2013.07.054, 2013.